# Topographic gradients of intrinsic dynamics across neocortex

**Golia Shafiei[1]\*, Ross D Markello[1], Reinder Vos de Wael[1], Boris C Bernhardt[1], Ben D Fulcher[2], Bratislav Misic[1]\***

[1]McConnell Brain Imaging Centre, Montréal Neurological Institute, McGill University, Montréal, Canada; [2]School of Physics, The University of Sydney, Sydney, Australia

**Abstract** The intrinsic dynamics of neuronal populations are shaped by both microscale attributes and macroscale connectome architecture. Here we comprehensively characterize the rich temporal patterns of neural activity throughout the human brain. Applying massive temporal feature extraction to regional haemodynamic activity, we systematically estimate over 6000 statistical properties of individual brain regions' time-series across the neocortex. We identify two robust spatial gradients of intrinsic dynamics, one spanning a ventromedial-dorsolateral axis and dominated by measures of signal autocorrelation, and the other spanning a unimodal-transmodal axis and dominated by measures of dynamic range. These gradients reflect spatial patterns of gene expression, intracortical myelin and cortical thickness, as well as structural and functional network embedding. Importantly, these gradients are correlated with patterns of meta-analytic functional activation, differentiating cognitive *versus* affective processing and sensory *versus* higher-order cognitive processing. Altogether, these findings demonstrate a link between microscale and macroscale architecture, intrinsic dynamics, and cognition.

\*For correspondence:
golia.shafiei@mail.mcgill.ca (GS);
bratislav.misic@mcgill.ca (BM)

## Introduction

The brain is a complex network of anatomically connected and perpetually interacting neuronal populations (*Sporns et al., 2005*). Inter-regional connectivity promotes signaling via electrical impulses, generating patterned electrophysiological and haemodynamic activity (*Avena-Koenigsberger et al., 2017*; *Suárez et al., 2020*). Neuronal populations are organized into a hierarchy of increasingly polyfunctional neural circuits (*Jones and Powell, 1970*; *Mesulam, 1998*; *Hilgetag and Goulas, 2020*; *Bazinet et al., 2020*), manifesting as topographic gradients of molecular and cellular properties that smoothly vary between unimodal and transmodal cortices (*Huntenburg et al., 2018*). Recent studies have demonstrated cortical gradients of gene transcription (*Fulcher et al., 2019*; *Burt et al., 2018*), intracortical myelin (*Huntenburg et al., 2017*), cortical thickness (*Wagstyl et al., 2015*) and laminar profiles (*Paquola et al., 2019*).

The topological and physical embedding of neural circuits in macroscale networks and microscale gradients influence their dynamics (*Kiebel et al., 2008*; *Gollo et al., 2015*; *Wang, 2020*). For a neuronal population, the confluence of local micro-architectural properties and global connectivity shapes both the generation of local rhythms, as well as its propensity to communicate with other populations. Specifically, cell type composition, their morphology and their configuration in local circuits determine how signals are generated, transmitted and integrated (*Payeur et al., 2019*). These micro-architectural properties – increasingly measured directly from histology or inferred from other measurements, such as microarray gene expression – provide a unique opportunity to relate circuit architecture to temporal dynamics and computation. Indeed, multiple studies have focused on how intrinsic timescales vary in relation to microscale and macroscale attributes (*Murray et al., 2014*; *Mahjoory et al., 2019*; *Shine et al., 2019*; *Gao et al., 2020*; *Ito et al., 2020*; *Raut et al., 2020*).

The primary functional consequence of this hierarchy of timescales is thought to be a hierarchy of temporal receptive windows: time windows in which a newly arriving stimulus will modify processing of previously presented (i.e. contextual) information (*Hasson et al., 2008*; *Honey et al., 2012*; *Baldassano et al., 2017*; *Huntenburg et al., 2018*; *Chaudhuri et al., 2015*; *Chien and Honey, 2020*). Thus, areas at the bottom of the hierarchy preferentially respond to immediate changes in the sensory environment, while responses in areas at the top of the hierarchy are modulated by prior context. Altogether, previous work highlights a hierarchy of a small number of manually selected time-series features, but it is possible that different types of local computations manifest as different organizational gradients.

The relationship between structure and dynamics is also observed at the network level (*Suárez et al., 2020*). Intrinsic or 'resting state' networks possess unique spectral fingerprints (*Keitel and Gross, 2016*). The signal variability of brain areas, measured in terms of standard deviations or temporal entropy, is closely related to their structural and functional connectivity profiles (i.e. network embedding) (*Mišić et al., 2011*; *Burzynska et al., 2013*; *Garrett et al., 2017*; *Shafiei et al., 2019*). More generally, the autocorrelation of blood oxygenation level-dependent (BOLD) signal is correlated with topological characteristics of structural brain networks, such that areas with greater connectivity generate signals with greater autocorrelation (*Sethi et al., 2017*; *Fallon et al., 2020*). Finally, in computational models of structurally coupled neuronal populations (neural mass and neural field models [*Breakspear, 2017*]), highly interconnected hubs exhibit slower dynamic fluctuations, while sensory areas exhibit fast fluctuating neural activity (*Gollo et al., 2015*). Indeed, these models offer better fits to empirical functional connectivity if they assume heterogeneous local dynamics (*Cocchi et al., 2016*; *Demirtaş et al., 2019*; *Wang et al., 2019*; *Deco et al., 2020*).

Altogether, multiple lines of evidence suggest that local computations may reflect systematic variation in microscale properties and macroscale network embedding, manifesting as diverse time-series features of regional neural activity. How molecular, cellular and connectomic architecture precisely shapes temporal dynamics, and ultimately, cortical patterns of functional specialization, is poorly understood. A significant limitation is that conventional computational analysis is based on specific, manually selected time-series features, such as the decay of the autocorrelation function, bands of the Fourier power spectrum, or signal variance. Yet the time-series analysis literature is vast and interdisciplinary; how do other metrics of temporal structure vary across the brain and what can they tell us about cortical organization?

Here we comprehensively chart summary features of spontaneous BOLD signals across the cerebral cortex (hereafter referred to as 'intrinsic dynamics'), mapping temporal organization to structural organization. We apply massive temporal feature extraction to resting state BOLD signals to derive a near-exhaustive time-series profile for each brain region. We then systematically investigate the relationship between local time-series features and gene expression, microstructure, morphology, structural connectivity and functional connectivity. Finally, we map time-series features to a meta-analytic atlas of cognitive ontologies to investigate how temporal dynamics shape regional functional specialization. We show that intrinsic dynamics reflect molecular and cytoarchitectonic gradients, as well as patterns of structural and functional connectivity. These spatial variations in intrinsic dynamics ultimately manifest as patterns of distinct psychological functions.

## Results

All analyses were performed on four resting state fMRI runs from the Human Connectome Project (*Van Essen et al., 2013*). The data were pseudorandomly divided into two samples of unrelated participants to form *Discovery* and *Validation* samples with $n = 201$ and $n = 127$, respectively (*Vos de Wael et al., 2018*). External replication was then performed using data from the Midnight Scan Club (*Gordon et al., 2017*). Massive temporal feature extraction was performed using highly comparative time-series analysis, *hctsa* (*Fulcher et al., 2013*; *Fulcher and Jones, 2017*), yielding 6441 features per regional time-series, including measures of frequency composition, variance, autocorrelation, fractal scaling and entropy (*Figure 1*). The results are organized as follows. We first investigate whether regions that are structurally and functionally connected display similar intrinsic dynamics. We then characterize the topographic organization of time-series features in relation to microstructural attributes and cognitive ontologies.

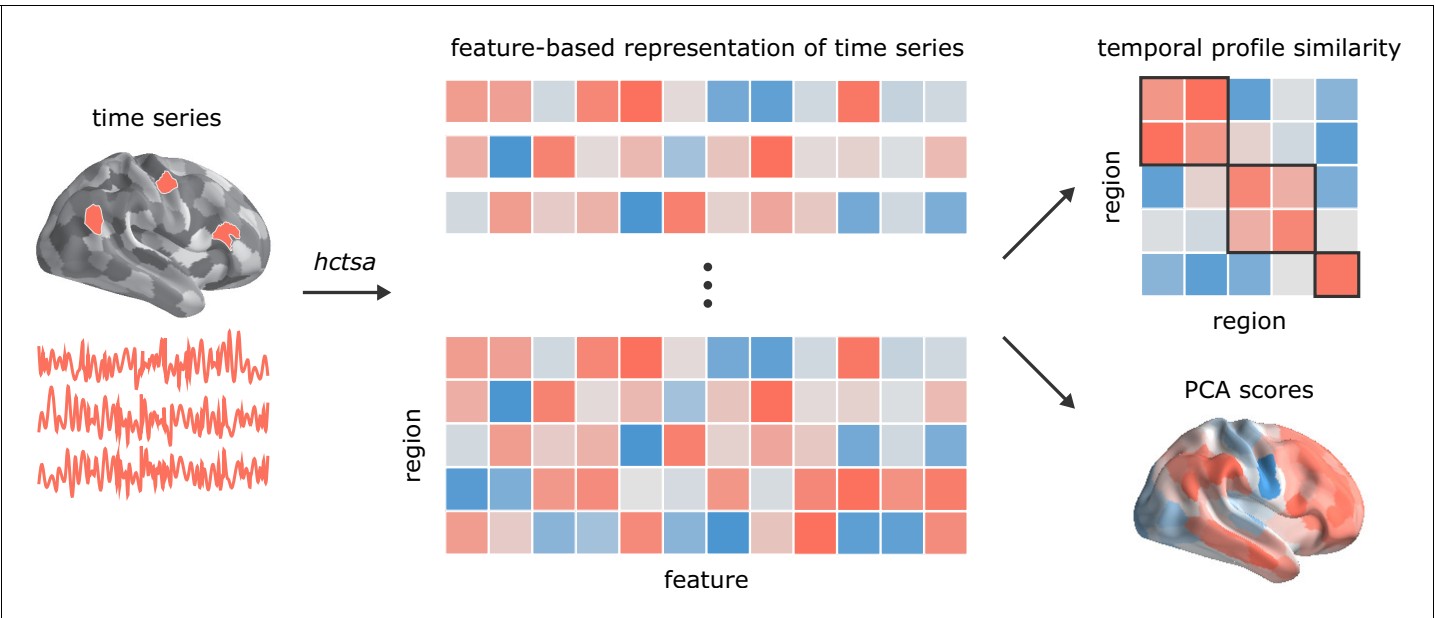

**Figure 1.** Temporal phenotyping of regional dynamics. The highly comparative time-series analysis toolbox, *hctsa* (*Fulcher et al., 2013*; *Fulcher and Jones, 2017*), was used to extract 6441 time-series features of the parcellated time-series for each brain region and participant, including measures of autocorrelation, variance, spectral power, entropy, etc. Regional time-series profiles were then entered into two types of analyses. In the first analysis, pairs of regional time-series feature vectors were correlated to generate a region × region temporal profile similarity network. In the second analysis, principal component analysis (PCA) was performed to identify orthogonal linear combinations of time-series features that vary maximally across the cortex.

## Inter-regional temporal profile similarity reflects network geometry and topology

We first assessed the extent to which intrinsic dynamics depend on inter-regional physical distance, anatomical connectivity and functional connectivity. We estimated similarity between inter-regional dynamics by computing Pearson correlation coefficients between regional time-series feature vectors (*Figure 1*). Two regional time-series are judged to be similar if they have similar temporal profiles, estimated across a comprehensive and diverse set of time-series features (e.g. similar entropy, stationarity, linear correlation properties) (*Fulcher, 2018*). This measure of similarity identifies pairs of regions that have similar dynamical features, but not necessarily coherent or synchronous dynamics (*Figure 2a*). We refer to correlations between regional time-series feature profiles as 'temporal profile similarity'.

*Figure 2b* shows a negative exponential relationship between spatial proximity and temporal profile similarity, meaning that regions that are spatially close exhibit similar intrinsic dynamics. Interestingly, regions that share an anatomical projection have greater temporal profile similarity than those that do not (*Figure 2c*; two-tailed t-test; $t(79,798) = 40.234$, $p \approx 0$). To test whether this anatomically-mediated similarity of time-series features is not due to spatial proximity, we performed two additional comparisons. First, we regressed out the exponential trend identified above from the temporal profile similarity matrix, and repeated the analysis on the residuals, yielding a significant difference in temporal profile similarity between connected and non-connected regions (two-tailed t-test; $t(79,798) = 9.916$, $p \approx 0$). Second, we generated an ensemble of 10,000 degree- and edge length-preserving surrogate networks (*Betzel and Bassett, 2018*), and compared the difference of the means between connected and non-connected pairs in the empirical and surrogate networks. Again, we observe a significant difference in temporal profile similarity between connected and non-connected regions (two-tailed; $p_{\mathrm{rewired}} = 0.0001$; *Figure 2c*).

Likewise, regions belonging to the same intrinsic functional network have greater temporal profile similarity compared to regions in different networks (*Figure 2d*; two-tailed t-test; $t(79,798) = 61.093$, $p \approx 0$). To confirm this finding is not driven by spatial proximity, we repeated the analysis with distance-residualized values (*Mišić et al., 2014*), finding a significant difference (two-tailed t-test;

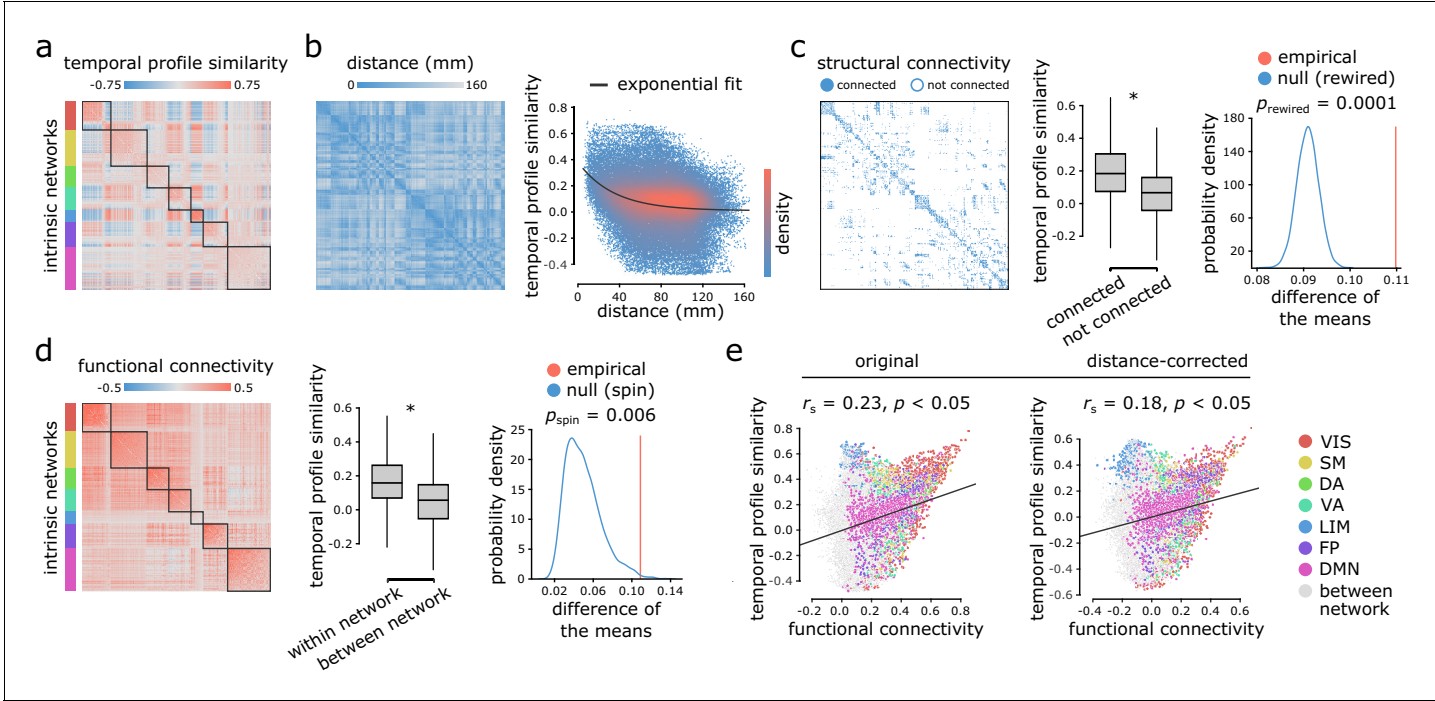

**Figure 2.** Inter-regional temporal profile similarity reflects network geometry and topology. (a) Temporal profile similarity networks are constructed by correlating pairs of regional time-series feature vectors. Brain regions are ordered based on their intrinsic functional network assignments (*Yeo et al., 2011*; *Schaefer et al., 2018*). (b) Temporal profile similarity between regions significantly decreases as a function of Euclidean distance between them. The black line represents an exponential fit as $y = 0.37e^{-0.03x} + 0.01$, where $y$ is temporal profile similarity and $x$ is Euclidean distance. (c, d) Regional time-series features are compared between pairs of cortical areas using their structural and functional connectivity profiles. Pairwise temporal profile similarity is significantly higher among structurally-connected areas (c), and among regions that belong to the same intrinsic functional networks (d). Asterisks denote a statistically significant difference of the means (two-tailed $t$-test; $p \approx 0$). For structural networks, statistical significance of the difference of the mean temporal profile similarity of connected and unconnected node pairs is also assessed against a null distribution of differences generated from a population degree- and edge length-preserving rewired networks (*Betzel and Bassett, 2018*) (c, right-most panel). For functional networks, statistical significance of the difference of the mean temporal profile similarity of within and between intrinsic networks is also assessed against a null distribution of differences generated by spatial autocorrelation-preserving label permutation ('spin tests'; *Alexander-Bloch et al., 2018*) (d, right-most panel). (e) Temporal profile similarity is positively correlated with functional connectivity. This relationship remains after partialling out Euclidean distance between regions from both measures using exponential trends. $r_s$ denotes the Spearman rank correlation coefficient; linear regression lines are added to the scatter plots for visualization purposes only. Connections are color-coded based on the intrinsic network assignments (*Yeo et al., 2011*; *Schaefer et al., 2018*). VIS = visual, SM = somatomotor, DA = dorsal attention, VA = ventral attention, LIM = limbic, FP = fronto parietal, DMN = default mode.

The online version of this article includes the following figure supplement(s) for figure 2:

**Figure supplement 1.** Intrinsic networks: 17 network partition.

**Figure supplement 2.** Functional connectivity measured by partial correlations.

$t(79, 798) = 47.112$, $p \approx 0$). We also repeated the analysis using a nonparametric label-permutation null model with preserved spatial autocorrelation (10,000 repetitions) (*Alexander-Bloch et al., 2018*; *Markello and Misic, 2020*), again finding significantly greater within- compared to between-network temporal profile similarity (two-tailed; $p_{spin} = 0.006$; *Figure 2d*). These results are consistent when applying the 17 network partition of intrinsic networks (*Yeo et al., 2011*; *Schaefer et al., 2018*; *Figure 2—figure supplement 1*).

More generally, we find a weak positive correlation between temporal profile similarity and functional connectivity (original: Spearman rank $r_s = 0.23$, $p \approx 0$; distance-corrected: $r_s = 0.18$, $p \approx 0$; *Figure 2e*), suggesting that areas with similar time-series features exhibit coherent spontaneous fluctuations, but that the two are only weakly correlated. *Figure 2e* shows the correlation between temporal profile similarity and functional connectivity; points represent node pairs and are colored by their membership in intrinsic networks (*Yeo et al., 2011*; *Schaefer et al., 2018*). The results are

consistent when functional connectivity is estimated using partial correlations (*Figure 2—figure supplement 2*). In other words, two regions could display similar time-series features, but they do not necessarily fluctuate coherently. Thus, representing time-series using sets of features provides a fundamentally different perspective compared to representing them as the raw set of ordered BOLD measurements.

As a final step, we sought to assess the distinct contributions of Euclidean distance, structural connectivity and functional connectivity to temporal profile similarity. Dominance analysis revealed the relative importance of each predictor (collective $R^2$ = 0.28; distance = 56%, structural connectivity = 20.4%, functional connectivity = 23.6%; *Supplementary file 1*), suggesting that distance contributes the most to temporal profile similarity, while structural and functional connectivity make distinct but approximately even contributions (*Budescu, 1993*; *Azen and Budescu, 2003*) (https://github.com/dominance-analysis/dominance-analysis). Altogether, we find that the organization of intrinsic dynamics is closely related to both the geometric and topological embedding of brain regions in macroscale networks.

## Two distinct spatial gradients of intrinsic dynamics

We next investigate the topographic organization of time-series features. The *hctsa* library generates 6 441 time-series features, with the aim of being comprehensive in coverage across scientific time-series analysis algorithms and, as a result, contains groups of features with correlated outputs (*Fulcher et al., 2013*). We therefore sought to identify groups of correlated features that explain maximal variance and that span different conceptual types of time-series properties. Applying principal component analysis (PCA; *Scikit-learn* [*Pedregosa et al., 2011*]) to the region × feature matrix yielded mutually orthogonal patterns of intrinsic dynamics (*Figure 1*), with the top two components collectively accounting for more than 70% of the variance in time-series features (*Figure 3a*). *Figure 3a* shows the spatial distribution of the top two components. The first component (PC1) mainly captures differential intrinsic dynamics along a ventromedial-dorsolateral gradient, separating occipital-parietal cortex and anterior temporal cortex. The second component (PC2) captures a unimodal-transmodal gradient, reminiscent of recently reported miscrostructural and functional gradients (*Huntenburg et al., 2018*). Both components show considerable hemispheric symmetry. In the following sections, we focus on these two components because of their (a) effect size (percent variance accounted for), (b) close resemblance to previously reported topographic gradients, and (c) reproducibility (only the first two components were reproducible in both the HCP and MSC datasets; see *Sensitivity and replication analyses* below). Note that neither spatial maps were significantly correlated with temporal signal-to-noise ratio map, computed as the ratio of the time-series mean to standard deviation (tSNR; PC1: $r_s = 0.28$, $p_{spin} = 0.19$; PC2: $r_s = 0.21$, $p_{spin} = 0.16$).

Which time-series features contribute most to these topographic gradients of intrinsic dynamics? To address this question, we systematically assess the feature composition of PC1 and PC2. We compute univariate correlations (i.e. loadings) between individual time-series feature vectors and PC scores (*Figure 3b*). Each loading is assessed against 10 000 spin tests and the results are corrected for multiple comparisons by controlling the false discovery rate (FDR [*Benjamini and Hochberg, 1995*]; $\alpha = 0.001$). The top 5% positively and negatively correlated features are shown in word clouds. The complete list of features (ranked by loading), their definitions, loadings and *p*-values for both components is presented in machine-readable format in *Supplementary files 3* and *4*. Altogether, we find that PC1 is sensitive to temporal dependencies in BOLD signals, while PC2 is sensitive to the distribution shape of time-series amplitudes. For PC1, in line with previous reports, we observe strong contributions from multiple measures of autocorrelation (e.g. linear autocorrelation; nonlinear autocorrelation; automutual information). Short-lag autocorrelation measures load positively, while long-lag autocorrelation measures load negatively, consistent with the notion that autocorrelation decays with increasing time lag (*Murray et al., 2014*; *Gao et al., 2020*; *Raut et al., 2020*; *Figure 3—figure supplement 1*). For PC2, we observe strong contributions from measures of distribution shape, captured by measures of distributional entropy (e.g. entropy of kernel-smoothed distribution; kurtosis; distribution balance about the mean). In other words, PC2 captures the spread of time-series amplitudes away from the mean. Interestingly, none of the odd moments (distribution asymmetry) are high in the PC2 loading list, just even moments, suggesting that PC2 captures the shape of the deviations of time-series data points in both directions from the mean. Thus, PC2

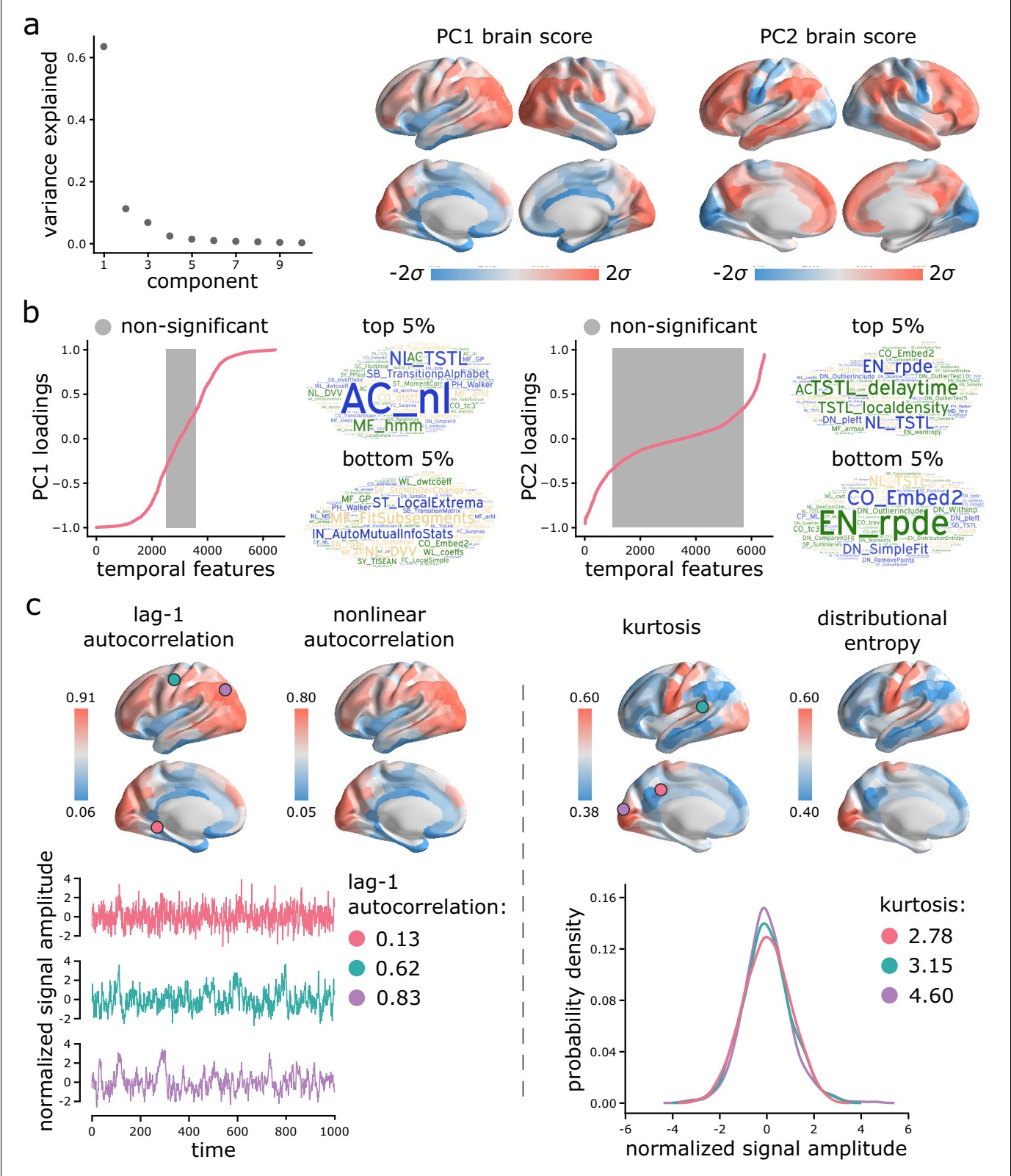

**Figure 3.** Topographic gradients of intrinsic dynamics. (**a**) PCA analysis identified linear combinations of hctsa time-series features with maximum variance across the cortex. Collectively, the first two components (PC1 and PC2) account for 75% of the total variance in time-series features of BOLD dynamics. To estimate the extent to which cortical regions display the patterns of intrinsic dynamics captured by each component, *hctsa* matrices were projected back onto the PC weights (eigenvectors), yielding spatial maps of brain scores for each component. Spatial maps are depicted based on the

*Figure 3 continued*

standard deviation $\sigma$ of their respective brain score distributions. (b) To understand the feature composition of the intrinsic dynamic patterns captured by PC1 and PC2, feature loadings were computed by correlating individual *hctsa* feature vectors with the PC score maps. PC loadings thus estimate the shared spatial variance between an individual time-series feature and the composite intrinsic dynamic map captured by a PC. time-series features are ordered by their individual loadings. Grey indicates non-significance based on 10,000 spatial permutation tests (FDR correction, $\alpha = 0.001$). Features corresponding to the top and bottom 5% of PC1 and PC2 are visualized using word clouds. The complete list of features (ranked by loading), their definitions, correlations and *p*-values for both components is presented in machine-readable format in *Supplementary files 3* and *4*. Feature nomenclature in *hctsa* is organized such that the term prefix indicates the broad class of measures (e.g. AC = autocorrelation, DN = distribution) and the term suffix indicates the specific measure (for a complete list, see https://hctsa-users.gitbook.io/hctsa-manual/list-of-included-code-files). (c) The spatial distributions of two high-loading representative time-series features are depicted for each component, including lag-1 linear autocorrelation (AC_1) and lag-[0,2,3] nonlinear autocorrelation (AC_nl_023, estimated as average $\langle x_t^2 x_{t-2} x_{t-3} \rangle$ across time-series $x$) for PC1; and kurtosis (DN_Moments_4) and entropy (EN_DistributionEntropy_ks__02) of the time-series points distribution for PC2. To build intuition about what each component reflects about regional signals, three regional time-series from one participant are selected based on their lag-1 autocorrelation and kurtosis (circles on the brain surface: pink = 5th percentile, green = 50th percentile, purple = 95th percentile). Going from low-ranked to high-ranked regions results in a slowing down of BOLD fluctuations for the former and increasingly heavier symmetric tails of the signal amplitude distributions for the latter. Note that normalized feature values are shown in the first row, whereas the raw feature values are shown in the second row.

The online version of this article includes the following figure supplement(s) for figure 3:

**Figure supplement 1.** Linear autocorrelation function.

indexes the range or diversity of values that a regional time-series can realize. Hereafter, we refer to the time-series profile of PC1 as 'autocorrelation' and PC2 as 'dynamic range'.

To illustrate the spatial organization and time-series attributes of these components, *Figure 3c* shows the spatial distributions of two high-loading representative time-series features for each component. Ventromedial areas (lower in the PC1 gradient) have lower linear and nonlinear autocorrelation, while doroslateral areas (higher in the PC1 gradient) have greater autocorrelation. Sensory areas (lower in the PC2 gradient) have greater distributional entropy and kurtosis, while transmodal areas (higher in the PC2 gradient) have lower distributional entropy and kurtosis. Finally, to build intuition about what each component reflects about regional signals, we select three regional time-series from one participant based on their lag-1 autocorrelation and kurtosis (*Figure 3c*; pink = 5th percentile, green = 50th percentile, purple = 95th percentile). For the former, going from low-ranked to high-ranked regions results in a slowing down of BOLD fluctuations. For the latter, going from low-ranked to high-ranked regions results in increasingly heavier symmetric tails of the signal amplitude distributions.

## Intrinsic dynamics reflect microscale and macroscale hierarchies

To assess whether the dominant variation in time-series features of BOLD dynamics varies spatially with structural and functional gradients, we next quantify the concordance between PC1/PC2 and multiple microstructural and functional attributes (*Figure 4*). Specifically, we compare PC1 and PC2 with the following microscale and macroscale features: (1) the first component of microarray gene expression computed from the Allen Institute Human Brain Atlas (*Hawrylycz et al., 2012*; *Burt et al., 2018*) using PCA analysis, (2) the principal gradient of functional connectivity estimated using diffusion map embedding (*Margulies et al., 2016*; *Coifman et al., 2005*; *Langs et al., 2015*) (https://github.com/satra/mapalign), (3) T1w/T2w ratio, a putative proxy for intracortical myelin (*Huntenburg et al., 2017*), (4) cortical thickness (*Wagstyl et al., 2015*). We use Spearman rank correlations ($r_s$) throughout, as they do not assume a linear relationship among variables. Given the spatially autocorrelated nature of both *hctsa* features and other imaging features, we assess statistical significance with respect to nonparametric spatial autocorrelation-preserving null models (*Alexander-Bloch et al., 2018*; *Markello and Misic, 2020*).

PC1 topography is correlated with the first principal component of gene expression ($r_s = 0.57$, $p_{spin} = 0.03$), but no other attributes. PC2 topography is significantly correlated with the first principal component of gene expression ($r_s = -0.45$, $p_{spin} = 0.0008$), with the principal gradient of functional connectivity ($r_s = 0.77$, $p_{spin} = 0.0001$), with T1w/T2w ratio ($r_s = -0.57$, $p_{spin} = 0.0001$), and with cortical thickness ($r_s = 0.43$, $p_{spin} = 0.008$). Altogether, the two topographic gradients of intrinsic dynamics closely mirror molecular and microstructural gradients, suggesting a link between regional structural properties and regional dynamical properties. *Figure 4—figure supplement 1* further confirms this

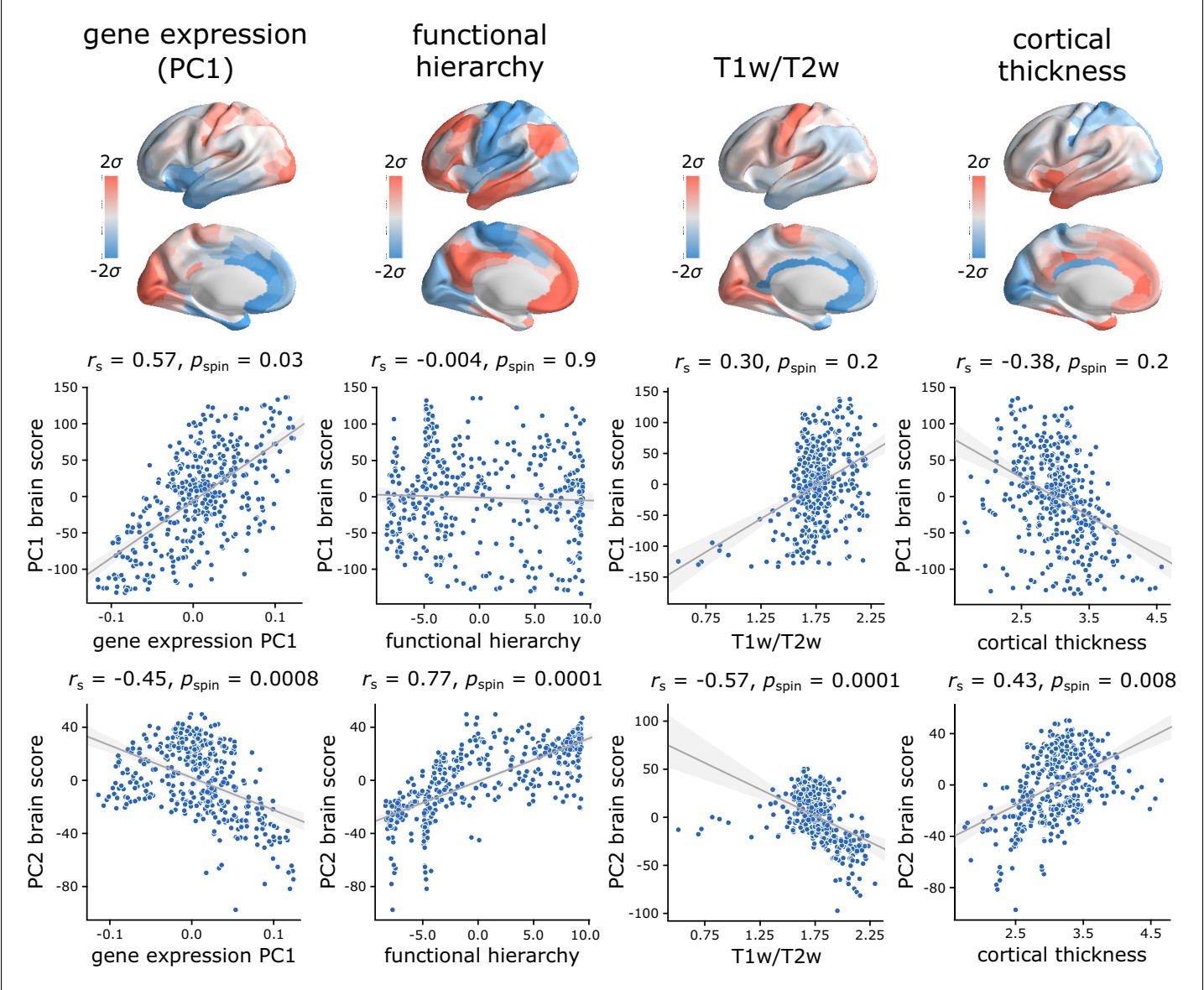

**Figure 4.** Hierarchical organization of intrinsic dynamics. PC1 and PC2 brain score patterns are compared with four molecular, microstructural and functional maps. These maps include the first principal component of microarray gene expression data from the Allen Human Brain Atlas (*Hawrylycz et al., 2012*; *Burt et al., 2018*), the first (principal) gradient of functional connectivity estimated using diffusion map embedding (*Margulies et al., 2016*; *Coifman et al., 2005*; *Langs et al., 2015*), group-average T1w/T2w ratio, and group-average cortical thickness. The three latter indices were computed from the HCP dataset (*Van Essen et al., 2013*). Statistical significance of the reported Spearman rank correlation $r_s$ is assessed using 10,000 spatial permutations tests, preserving the spatial autocorrelation in the data ('spin tests'; *Alexander-Bloch et al., 2018*). Linear regression lines are added to the scatter plots for visualization purposes only.

The online version of this article includes the following figure supplement(s) for figure 4:

**Figure supplement 1.** Intrinsic dynamics across intrinsic networks, cytoarchitectonic classes and laminar differentiation levels.

**Figure supplement 2.** Topographic organization of intrinsic dynamics compared to evolutionary expansion and participation coefficient.

intuition, showing the mean score of each component for three well-known cortical partitions, including intrinsic functional networks (*Yeo et al., 2011*; *Schaefer et al., 2018*), cytoarchitectonic classes (*von Economo and Koskinas, 1925*; *von Economo et al., 2008*; *Vértes et al., 2016*) and laminar differentiation levels (*Mesulam, 2000*).

For completeness, we also tested associations with two maps that were previously related to cortical hierarchies: evolutionary expansion (indexing enlargement of cortical areas in the human relative

to the macaque) (*Hill et al., 2010*; *Baum et al., 2020*) and node-wise functional participation coefficient (indexing the diversity of a node's links) (*Bertolero et al., 2017*; *Baum et al., 2020*). PC2 is significantly correlated with evolutionary expansion ($r_s = 0.52$, $p_{spin} = 0.0002$), but neither component is correlated with participation coefficient (*Figure 4—figure supplement 2*).

## Spatial gradients of intrinsic dynamics support distinct functional activations

Given that topographic patterns of intrinsic dynamics run parallel to microstructural and functional gradients, and are marked by specific time-series features, we next asked whether these topographic patterns of intrinsic dynamics are related to patterns of functional activation and psychological processes. To address this question, we used Neurosynth to derive probability maps for multiple psychological terms (*Yarkoni et al., 2011*). The term set was restricted to those in the intersection of terms reported in Neurosynth and in the Cognitive Atlas (*Poldrack et al., 2011*), yielding a total of 123 terms (*Supplementary file 2*). Each term map was correlated with the PC1 and PC2 score maps to identify topographic distributions of psychological terms that most closely correspond to patterns of intrinsic dynamics (Bonferroni corrected, $\alpha = 0.05$; *Figure 5*). Consistent with the intuition developed from comparisons with intrinsic networks, PC1 intrinsic dynamics mainly defined a cognitive-affective axis (e.g. 'attention' *versus* 'stress', 'fear', 'loss', 'emotion'; *Figure 5a*), while PC2 dynamics defined a sensory-cognitive axis (e.g. 'perception', 'multisensory', 'facial expression' *versus* 'cognitive control', 'memory retrieval', 'reasoning'; *Figure 5b*).

## Sensitivity and replication analyses

As a final step, we sought to assess the extent to which the present findings are replicable under alternative processing choices and in other samples (*Figure 6*). For all comparisons, we correlated PC1 and PC2 scores and weights obtained in the original analysis and in each new analysis. Significance was assessed using spatial autocorrelation preserving nulls as before. We first replicated the results in individual subjects in the *Discovery* sample by applying PCA to individual region × feature matrices and aligning PCA results through an iterative process using Procrustes rotations (https://github.com/satra/mapalign [*Langs et al., 2015*]). The mean individual-level PC scores and weights were then compared to the original findings (*Figure 6a*). We next replicated the results by repeating the analysis after grey-matter signal regression (similar to global signal regression as the global signal is shown to be a grey-matter specific signal following sICA+FIX) (*Glasser et al., 2018*; *Glasser et al., 2016*), with near identical results (*Figure 6b*). To assess the extent to which results are influenced by choice of parcellation, we repeated the analysis using the 68-region Desikan-Killiany anatomical atlas (*Desikan et al., 2006*), which were then further divided into 200 approximately equally-sized cortical areas. Again, we find near-identical results (*Figure 6c*).

In the last two analyses, we focused on out-of-sample validation. We first repeated the analysis on the held-out *Validation* sample of $n = 127$ unrelated HCP subjects, with similar results (*Figure 6d*). Finally, we repeated the analysis using data from the independently collected Midnight Scan Club (MSC) dataset, again finding highly consistent results (*Figure 6e*).

## Discussion

In the present report, we comprehensively characterize intrinsic dynamics across the cortex, identifying two robust spatial patterns of time-series features. The patterns, capturing spatial variation in signal autocorrelation and dynamic range, follow microscale gradients and macroscale network architecture. Importantly, the two patterns underlie distinct psychological axes, demonstrating a link between brain architecture, intrinsic dynamics, and cognition. These findings are robust against a wide range of methodological choices and were validated in two held-out samples.

Our results demonstrate that regional haemodynamic activity, often overlooked in favour of electrophysiological measurements with greater temporal resolution, possesses a rich dynamic signature (*Garrett et al., 2013*; *Uddin, 2020*; *Preti et al., 2017*; *Lurie et al., 2020*; *Li et al., 2019*; *Bolt et al., 2018*). While multiple reports have suggested the existence of a timescale or temporal receptive window hierarchy (*Kiebel et al., 2008*; *Murray et al., 2014*; *Honey et al., 2012*; *Hasson et al., 2008*; *Watanabe et al., 2019*; *Golesorkhi et al., 2020*; *Ito et al., 2020*), these investigations typically involved (a) incomplete spatial coverage, making it difficult to quantitatively assess

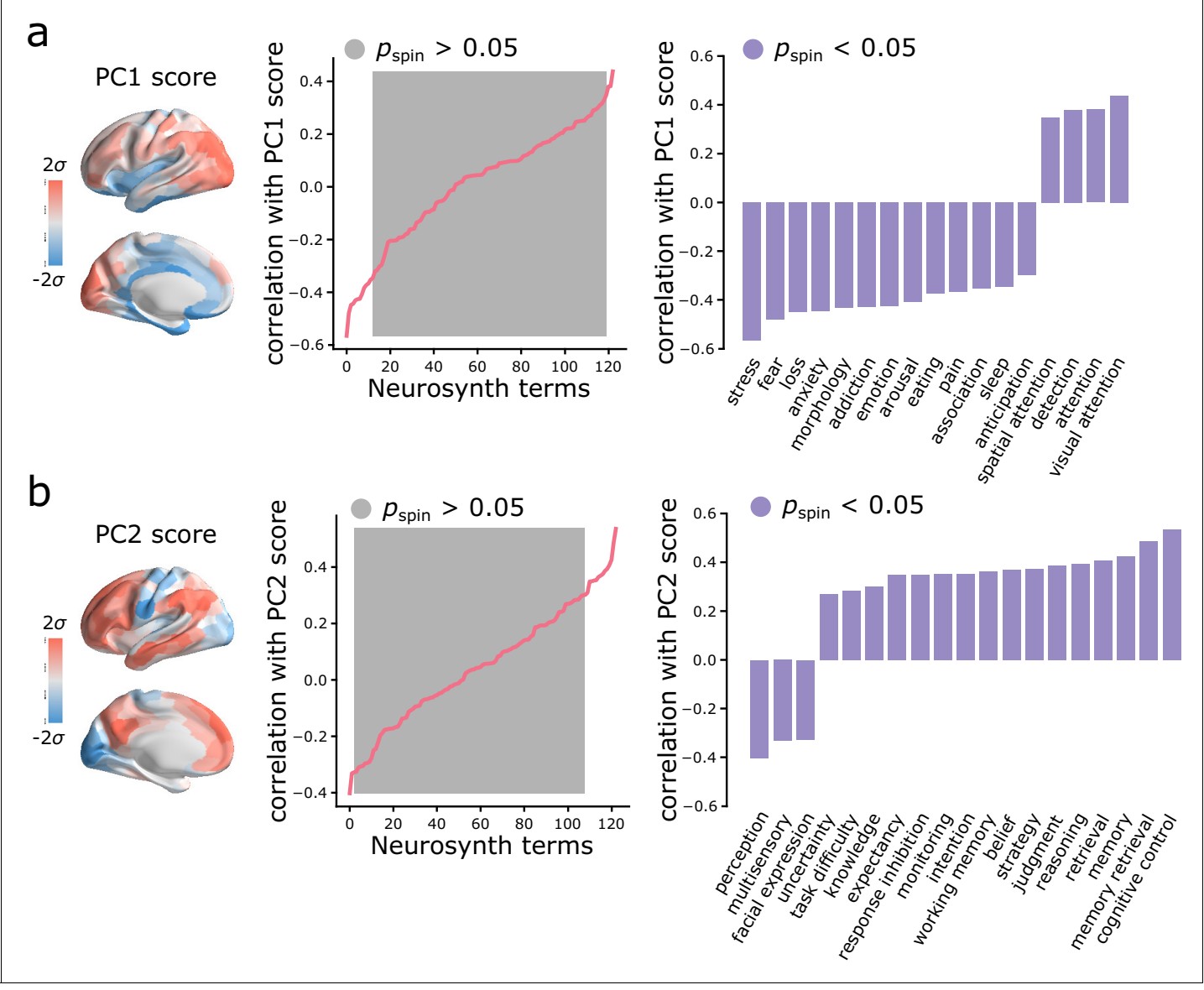

**Figure 5.** Spatial gradients of intrinsic dynamics support distinct functional activations. We used Neurosynth to derive probability maps for multiple psychological terms (*Yarkoni et al., 2011*). The term set was restricted to those in the intersection of terms reported in Neurosynth and in the Cognitive Atlas (*Poldrack et al., 2011*), yielding a total of 123 terms (*Supplementary file 2*). Each term map was correlated with the PC1 (a) and PC2 (b) score maps to identify topographic distributions of psychological terms that most closely correspond to patterns of intrinsic dynamics. Grey indicates non-significance based on 10,000 spatial permutation tests (Bonferroni correction, $\alpha = 0.05$). Statistically significant terms are shown on the right.

correspondence with other microscale and macroscale maps, and (b) a priori measures of interest, such as spectral power or temporal autocorrelation, potentially obscuring other important dynamical features. Here we comprehensively benchmark the entire dynamic profile of the brain, by near-exhaustively estimating 6000+ features from the wider time-series literature. We identify a much broader spectrum of time-series features that relate to microstructure, connectivity and behavior. As we discuss below, feature-based time-series phenotyping offers a powerful, fundamentally new and entirely data-driven method to quantify and articulate neural dynamics.

Applying a data-driven feature extraction method to high-resolution BOLD fMRI, we decompose regional signals into two intrinsic modes, with distinct topographic organization and time-series features. One pattern, characterized by variation in signal autocorrelation, follows a ventromedial-dorsolateral gradient, separating the limbic and paralimbic systems from posterior parietal cortex.

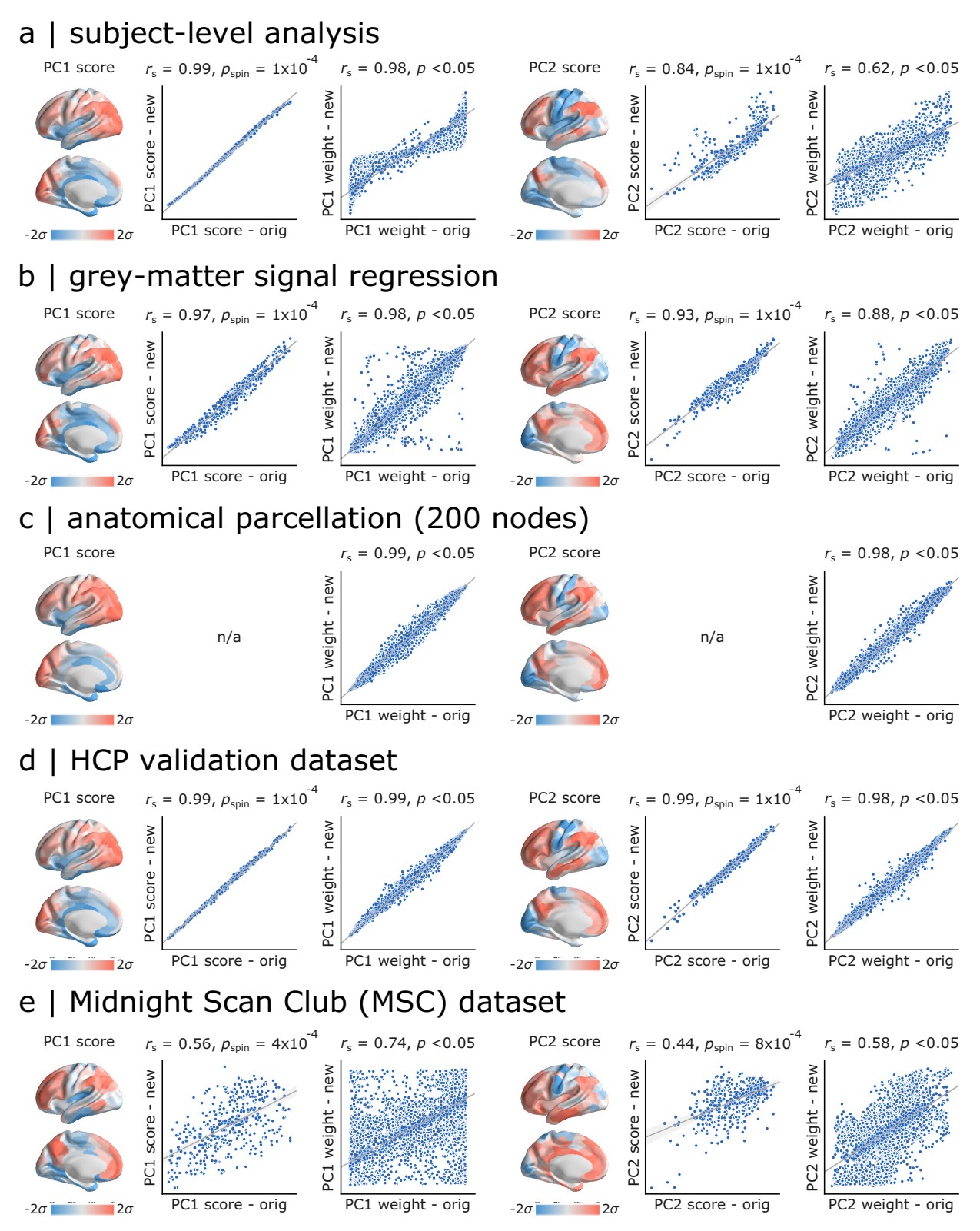

**Figure 6.** Sensitivity and replication analyses. For all comparisons, we correlated PC1 and PC2 scores and weights obtained in the original analysis and in each new analysis. Significance was assessed using spatial autocorrelation preserving nulls. Specific analyses include: (**a**) comparing group-level and individual subject-level results, (**b**) comparing data with and without grey-matter signal regression, (**c**) comparing functional (Schaefer) and anatomical parcellations (Desikan-Killiany), (**d**) comparing HCP *Discovery* and *Validation* datasets, (**e**) comparing HCP *Discovery* and Midnight Scan Club datasets.

Another pattern, characterized by dynamic range, follows a unimodal-transmodal gradient, separating primary sensory-motor cortices from association cortex. The first is closely associated with gene expression PC1 (itself closely related to cell type composition, synaptic physiology and cortical cytoarchitecture [*Burt et al., 2018*]), suggesting a molecular and cellular basis for regional differences in temporal autocorrelation. The second is closely associated with the principal functional gradient, as well as with intracortical myelin and cortical thickness, suggesting that the dynamic range of BOLD signals is related to regional variation in macroscale circuit organization. Taken together, we find evidence that molecular and cellular properties (gene expression PC1) relate to regional autocorrelation, while micro-circuit properties (T1w/T2w, cortical thickness) and macroscale network embedding (principal functional gradient) relate to regional dynamic range.

An emerging literature emphasizes the hierarchical organization of neural systems, whereby systematic variation in laminar architecture across the cortical sheet is mirrored by multiple cytological properties, including neuron density, spine count, branching and neurotransmitter receptor profiles (*Mesulam, 1998*; *Margulies et al., 2016*; *Hilgetag and Goulas, 2020*). These variations manifest as spatially ordered gradients of structural and functional attributes (*Huntenburg et al., 2018*), including gene expression (*Burt et al., 2018*; *Fulcher et al., 2019*; *Hansen et al., 2020*), cortical thickness (*Wagstyl et al., 2015*), intracortical myelin (*Huntenburg et al., 2017*), laminar differentiation (*Paquola et al., 2019*; *Wagstyl et al., 2020*) and excitability (*Demirtaş et al., 2019*; *Wang, 2020*; *Markicevic et al., 2020*; *Straub et al., 2020*). Indeed, we find that the two patterns of intrinsic dynamics are closely related to gene expression, intracortical myelin and cortical thickness. Our results build on this literature, demonstrating that microscale and connectional hierarchies leave an indelible mark on intrinsic dynamics (*Lurie and D'Esposito, 2020*), perhaps through variation in local excitability (*Demirtaş et al., 2019*; *Wang et al., 2019*; *Wang, 2020*; *Deco et al., 2020*). How these patterns are related to underlying cell types and subcortical afferent input – in particular, thalamo-cortical feedback – is an important ongoing question (*Abeysuriya et al., 2015*; *Garrett et al., 2018*; *Shine et al., 2019*; *Wang et al., 2019*; *Muller et al., 2020*; *Paquola et al., 2020*).

Importantly, the two patterns are related to two dominant axes of meta-analytic functional activation. We show that topographic variations in microcircuitry and connectomic embedding yield variations in intrinsic dynamics and may explain regional differences in functional specialization. The ventromedial-dorsolateral autocorrelation pattern differentiates affective versus cognitive activation (mainly visual cognition and visuo-spatial attention), whereas the unimodal-transmodal dynamic range pattern differentiates primary sensory versus higher-order cognitive processing. Collectively, these results provide evidence that local computations reflect systematic variation in multiple anatomical circuit properties, and can be measured as unique temporal signatures in regional activity and patterns of functional specialization.

More generally, the present findings are part of a larger trend in the field to understand structure-function relationships by considering molecular (*Richiardi et al., 2015*; *Fulcher and Fornito, 2016*; *Anderson et al., 2018*; *Zheng et al., 2019*), cellular (*Scholtens et al., 2014*; *Anderson et al., 2020*; *Shafiei et al., 2020*; *Muller et al., 2020*) and physiological (*Sethi et al., 2017*; *Fallon et al., 2020*) attributes of network nodes, thereby conceptually linking local and global brain organization (*Khambhati et al., 2018*; *Suárez et al., 2020*). In such 'annotated networks', macroscale network architecture is thought to reflect similarity in local properties, and *vice versa*, such that areas with similar properties are more likely to be anatomically connected and to functionally interact with one another (*Beul et al., 2017*; *Goulas et al., 2019*; *Wei et al., 2019*; *Hilgetag et al., 2019*). Indeed, we find that two regions are more likely to display similar intrinsic dynamics if they are anatomically connected and if they are part of the same functional community, suggesting that network organization and local intrinsic dynamics are intertwined (*Gollo et al., 2015*; *Cocchi et al., 2016*). A significant corollary of the present work is that functional connectivity – presently conceptualized as coherent fluctuations in neural activity and operationalized as correlated BOLD values over time – misses out on an important set of inter-regional relationships. Namely, two regions may display identical time-series profiles, suggesting common circuit dynamics and function, but unless they also display time-locked activity, current methods would miss out on this potentially biologically meaningful inter-regional relationship.

The present results are consistent with contemporary theories linking brain structure and function, but they must be interpreted with respect to several methodological caveats. First, all analyses were performed on BOLD time-series with lower sampling rate compared to electromagnetic recordings,

potentially obscuring more subtle dynamics occurring on faster timescales. To mitigate this concern, all analyses were performed in high-resolution multiband HCP data with multiple runs, and replicated in MSC data, but in principle, these analyses could be repeated and validated in magnetoencephalographic recordings (*Watanabe et al., 2019*). Second, all analyses were performed on haemodynamic time courses that may not completely reflect the underlying neuronal population dynamics. Despite this caveat, we observe a close correspondence between the isolated patterns of intrinsic dynamics and molecular, structural, functional, and psychological gradients. Third, the pattern of temporal signal-to-noise ratio in the BOLD is known to be non-uniform, but it is not correlated with the intrinsic dynamics patterns observed in the present report. Fourth, the analysis included all features from *hctsa*, potentially biasing results towards specific properties of BOLD signals. We attempted to mitigate this challenge by applying PCA to directly examine correlation patterns among features, but PCA components may still lend greater weight to over-represented feature classes (*Fulcher et al., 2013*). This may obscure the contribution of under-represented feature classes, and should be investigated further in future work.

Altogether, the present results point towards highly patterned intrinsic dynamics across the neocortex. These patterns reflect prominent molecular and microstructural gradients, as well as macroscale structural and functional organization. Importantly, spatial variation of intrinsic dynamics parallels spatial variation of meta-analytic cognitive functional activation. These findings demonstrate that structural organization of the brain shapes patterns of intrinsic dynamics, ultimately manifesting as distinct axes of psychological processes.

## Materials and methods

### Dataset: human connectome project (HCP)

Following the procedure described in *Vos de Wael et al., 2018*, we obtained structural and functional magnetic resonance imaging (MRI) data of two sets of healthy young adults (age range 22–35 years) with no familial relationships (neither within nor between sets) as *Discovery* ($n = 201$) and *Validation* ($n = 127$) sets from Human Connectome Project (HCP; S900 release [*Van Essen et al., 2013*]). All four resting state fMRI scans (two scans (R/L and L/R phase encoding directions) on day 1 and two scans (R/L and L/R phase encoding directions) on day 2, each about 15 min long; $TR = 720$ ms), as well as structural MRI and diffusion weighted imaging (DWI) data were available for all participants.

### HCP data processing

All the structural and functional MRI data were pre-processed using HCP minimal pre-processing pipelines (*Van Essen et al., 2013*; *Glasser et al., 2013*). We provide a brief description of data pre-processing below, while detailed information regarding data acquisition and pre-processing is available elsewhere (*Van Essen et al., 2013*; *Glasser et al., 2013*). The procedure was separately repeated for *Discovery* and *Validation* sets.

#### Structural MRI

T1- and T2- weighted MR images were corrected for gradient nonlinearity, and when available, the images were co-registered and averaged across repeated scans for each individual. The corrected T1w and T2w images were co-registered and cortical surfaces were extracted using FreeSurfer 5.3.0-HCP (*Vos de Wael et al., 2018*; *Dale et al., 1999*; *Fischl et al., 1999*). For each individual, cortical thickness was estimated as the difference between pial and white matter surfaces and T1w/T2w ratio was calculated as a putative proxy for intracortical myelin content. The pre-processed data were parcellated into 400 cortical areas using Schaefer parcellation (*Schaefer et al., 2018*).

#### Resting state functional MRI

All 3T functional MRI time-series were corrected for gradient nonlinearity, head motion using a rigid body transformation, and geometric distortions using scan pairs with opposite phase encoding directions (R/L, L/R) (*Vos de Wael et al., 2018*). Further pre-processing steps include co-registration of the corrected images to the T1w structural MR images, brain extraction, normalization of whole brain intensity, high-pass filtering (>2000s FWHM; to correct for scanner drifts), and removing

additional noise using the ICA-FIX process (*Vos de Wael et al., 2018*; *Salimi-Khorshidi et al., 2014*). The pre-processed time-series were then parcellated into 400 areas as described above. The parcellated time-series were used to construct functional connectivity matrices as a Pearson correlation coefficient between pairs of regional time-series for each of the four scans of each participant. A group-average functional connectivity matrix was constructed as the mean functional connectivity across all individuals and scans.

## Diffusion weighted imaging (DWI)

DWI data was pre-processed using the MRtrix3 package (*Tournier et al., 2019*) (https://www.mrtrix.org/). More specifically, fiber orientation distributions were generated using the multi-shell multi-tissue constrained spherical deconvolution algorithm from MRtrix (*Dhollander et al., 2016*; *Jeurissen et al., 2014*). White matter edges were then reconstructed using probabilistic streamline tractography based on the generated fiber orientation distributions (*Tournier et al., 2010*). The tract weights were then optimized by estimating an appropriate cross-section multiplier for each streamline following the procedure proposed by Smith and colleagues (*Smith et al., 2015*) and a connectivity matrix was built for each participant using the same parcellation as described above. Finally, we used a consensus approach to construct a binary group-level structural connectivity matrix, preserving the edge length distribution in individual participants (*Mišić et al., 2015*; *Betzel et al., 2019*; *Shafiei et al., 2020*; *Mišic et al., 2018*).

## Replication dataset: Midnight Scan Club (MSC)

We used resting state fMRI data of $n = 10$ healthy young adults, each with 10 scan sessions of about 30 min long, from Midnight Scan Club (MSC *Gordon et al., 2017*) dataset as an independent replication dataset. Details about the participants, MRI acquisition, and data pre-processing are provided by Gordon and colleagues elsewhere (*Gordon et al., 2017*). We obtained the surface-based, pre-processed resting state fMRI time courses in CIFTI format through OpenNeuro (https://openneuro.org/datasets/ds000224/versions/1.0.0). The pre-processing steps include motion correction and global signal regression (*Gordon et al., 2017*). Following the pre-processing methods suggested by *Gordon et al., 2017*, we smoothed the surface-level time-series data with geodesic 2D Gaussian kernels ($\sigma = 2.55$ mm) using the Connectome Workbench (*Marcus et al., 2011*). Finally, we censored the motion-contaminated frames of time-series for each participant separately, using the temporal masks provided with the dataset. The pre-processed data were parcellated into 400 cortical regions using Schaefer parcellation (*Schaefer et al., 2018*). One participant (MSC08) was excluded from subsequent analysis due to low data reliability and self-reported sleep as described in *Gordon et al., 2017*. The parcellated time-series were then subjected to the same analyses that were performed on the HCP *Discovery* and *Validation* datasets.

## Microarray expression data: Allen Human Brain Atlas (AHBA)

Regional microarray expression data were obtained from six post-mortem brains provided by the Allen Human Brain Atlas (AHBA; http://human.brain-map.org/) (*Hawrylycz et al., 2012*). We used the *abagen* (https://github.com/netneurolab/abagen; *Markello et al., 2020*) toolbox to process and map the data to 400 parcellated brain regions from Schaefer parcellation (*Schaefer et al., 2018*).

Briefly, genetic probes were reannotated using information provided by *Arnatkeviciute et al., 2019* instead of the default probe information from the AHBA dataset. Using reannotated information discards probes that cannot be reliably matched to genes. The reannotated probes were filtered based on their intensity relative to background noise levels (*Quackenbush, 2002*); probes with intensity less than background in $\geq 50\%$ of samples were discarded. A single probe with the highest differential stability, $\Delta_S(p)$, was selected to represent each gene (*Hawrylycz et al., 2015*), where differential stability was calculated as:

$$\Delta_S(p) = \frac{1}{\binom{N}{2}} \sum_{i=1}^{N-1} \sum_{j=i+1}^{N} \rho[B_i(p), B_j(p)] \tag{1}$$

Here, $\rho$ is Spearman's rank correlation of the expression of a single probe $p$ across regions in two

donor brains, $B_i$ and $B_j$, and $N$ is the total number of donor brains. This procedure retained 15,656 probes, each representing a unique gene.

Next, tissue samples were mirrored across left and right hemispheres (*Romero-Garcia et al., 2018*) and then assigned to brain regions using their corrected MNI coordinates (https://github.com/chrisfilo/alleninf) by finding the nearest region, up to 2 mm away. To reduce the potential for misassignment, sample-to-region matching was constrained by hemisphere and cortical/subcortical divisions (*Arnatkeviciute et al., 2019*). If a brain region was not assigned any sample based on the above procedure, the sample closest to the centroid of that region was selected in order to ensure that all brain regions were assigned a value. Samples assigned to the same brain region were averaged separately for each donor. Gene expression values were then normalized separately for each donor across regions using a robust sigmoid function and rescaled to the unit interval (*Fulcher and Fornito, 2016*). Scaled expression profiles were finally averaged across donors, resulting in a single matrix with rows corresponding to brain regions and columns corresponding to the retained 15,656 genes. The expression values of 1906 brain-specific genes were used for further analysis (*Burt et al., 2018*).

## Massive temporal feature extraction using *hctsa*

We used the highly comparative time-series analysis toolbox, *hctsa* (*Fulcher et al., 2013*; *Fulcher and Jones, 2017*), to perform a massive feature extraction of the time-series of each brain area for each participant. The *hctsa* package extracted over 7000 local time-series features using a wide range of operations based on time-series analysis (*Fulcher et al., 2013*; *Fulcher and Jones, 2017*). The extracted features include, but are not limited to, distributional properties, entropy and variability, autocorrelation, time-delay embeddings, and nonlinear properties of a given time-series (*Fulcher et al., 2013*; *Fulcher, 2018*).

The *hctsa* feature extraction analysis was performed on the parcellated fMRI time-series of each run and each participant separately (*Figure 1*). Following the feature extraction procedure, the outputs of the operations that produced errors were removed and the remaining features (6441 features) were normalized across nodes using an outlier-robust sigmoidal transform. We used Pearson correlation coefficients to measure the pairwise similarity between the time-series features of all possible combinations of brain areas. As a result, a temporal profile similarity network was constructed for each individual and each run, representing the strength of the similarity of the local temporal fingerprints of brain areas (*Figure 1*). The resulting similarity matrices were then compared to the underlying functional and structural brain networks.

## Neurosynth

Functional activation probability maps were obtained for multiple psychological terms using Neurosynth (*Yarkoni et al., 2011*) (https://github.com/neurosynth/neurosynth). Probability maps were restricted to those for terms present in both Neurosynth and the Cognitive Atlas (*Poldrack et al., 2011*), yielding a total of 123 maps (*Supplementary file 2*). We used the volumetric 'association test' (i.e. reverse inference) maps, which were projected to the FreeSurfer *fsaverage5* mid-grey surface with nearest neighbor interpolation using Freesurfer's *mri_vol2surf* function (v6.0.0; http://surfer.nmr.mgh.harvard.edu/). The resulting surface maps were then parcellated to 400 cortical regions using the Schaefer parcellation (*Schaefer et al., 2018*).

## Null model

A consistent question in the present work is the topographic correlation between time-series features and other features of interest. To make inferences about these links, we implement a null model that systematically disrupts the relationship between two topographic maps but preserves their spatial autocorrelation (*Alexander-Bloch et al., 2018*; *Markello and Misic, 2020*) (see also *Burt et al., 2018*; *Burt et al., 2020* for an alternative approach). We first created a surface-based representation of the Cammoun atlas on the FreeSurfer fsaverage surface using the Connectome Mapper toolkit (https://github.com/LTS5/cmp; *Daducci et al., 2012*). We used the spherical projection of the *fsaverage* surface to define spatial coordinates for each parcel by selecting the vertex closest to the center-of-mass of each parcel (*Vázquez-Rodríguez et al., 2019*; *Shafiei et al., 2020*; *Vézquez-Rodríguez et al., 2020*). The resulting spatial coordinates were used to generate null

models by applying randomly-sampled rotations and reassigning node values based on the closest resulting parcel (10,000 repetitions). The rotation was applied to one hemisphere and then mirrored to the other hemisphere.

## Acknowledgements

We thank Vincent Bazinet, Justine Hansen, Estefany Suarez, Bertha Vazquez-Rodriguez and Zhen-Qi Liu for helpful comments and stimulating discussion. This research was undertaken thanks in part to funding from the Canada First Research Excellence Fund, awarded to McGill University for the Healthy Brains for Healthy Lives initiative. BM acknowledges support from the Natural Sciences and Engineering Research Council of Canada (NSERC Discovery Grant RGPIN #017–04265) and from the Canada Research Chairs Program. GS acknowledges support from the Natural Sciences and Engineering Research Council of Canada (NSERC).

## Additional information

### Funding

| Funder | Grant reference number | Author |
|---|---|---|
| Natural Sciences and Engineering Research Council of Canada | | Golia Shafiei |
| Natural Sciences and Engineering Research Council of Canada | NSERC Discovery Grant RGPIN #017-04265 | Bratislav Misic |
| Canada First Research Excellence Fund | | Bratislav Misic |
| Canada Research Chairs | | Bratislav Misic |

The funders had no role in study design, data collection and interpretation, or the decision to submit the work for publication.

### Author contributions

Golia Shafiei, Conceptualization, Resources, Data curation, Formal analysis, Validation, Investigation, Visualization, Methodology, Writing - original draft, Project administration, Writing - review and editing; Ross D Markello, Data curation, Software, Methodology, Writing - review and editing; Reinder Vos de Wael, Boris C Bernhardt, Data curation, Writing - review and editing; Ben D Fulcher, Resources, Software, Methodology, Writing - review and editing; Bratislav Misic, Conceptualization, Supervision, Funding acquisition, Validation, Investigation, Visualization, Methodology, Writing - original draft, Project administration, Writing - review and editing

### Author ORCIDs

Golia Shafiei (iD) https://orcid.org/0000-0002-2036-5571
Ross D Markello (iD) http://orcid.org/0000-0003-1057-1336
Ben D Fulcher (iD) http://orcid.org/0000-0002-3003-4055
Bratislav Misic (iD) https://orcid.org/0000-0003-0307-2862

### Ethics

Human subjects: Informed consent and consent to publish were obtained during data acquisition process (all data used in this study were obtained from publicly available datasets).

### Decision letter and Author response

Decision letter https://doi.org/10.7554/eLife.62116.sa1
Author response https://doi.org/10.7554/eLife.62116.sa2

# Additional files

## Supplementary files

• Supplementary file 1. Dominance analysis. Dominance Analysis was used to quantify the distinct contributions of inter-regional Euclidean distance, structural connectivity, and functional connectivity to temporal profile similarity (*Budescu, 1993*; *Azen and Budescu, 2003*) (https://github.com/dominance-analysis/dominance-analysis). Dominance analysis is a method for assessing the relative importance of predictors in regression or classification models. The technique estimates the relative importance of predictors by constructing all possible combinations of predictors and quantifying the relative contribution of each predictor as additional variance explained (i.e. gain in $R^2$) by adding that predictor to the models. Specifically, for $p$ predictors we have $2^p - 1$ models that include all possible combinations of predictors. The incremental $R^2$ contribution of each predictor to a given subset model of all the other predictors is then calculated as the increase in $R^2$ due to the addition of that predictor to the regression model. Here we first constructed a multiple linear regression model with distance, structural connectivity and functional connectivity as independent variables and temporal profile similarity as the dependent variable to quantify the distinct contribution of each factor using dominance analysis. The total $R^2$ is 0.28 for the complete model that includes all variables. The relative importance of each factor is summarized in the table, where each column corresponds to: Interactional Dominance is the incremental $R^2$ contribution of the predictor to the complete model. For each variable, interactional dominance is measured as the difference between the $R^2$ of the complete model and the $R^2$ of the model with all other variables except that variable; Individual Dominance of a predictor is the $R^2$ of the model when only that predictor is included as the independent variable in the regression; Average Partial Dominance is the average incremental $R^2$ contributions of a given predictor to all possible subset of models except the complete model and the model that only includes that variable; Total Dominance is a summary measure that quantifies the additional contribution of each predictor to all subset models by averaging all the above measures for that predictor; Percentage Relative Importance is the percent value of the Total Dominance.

• Supplementary file 2. List of terms used in Neurosynth analyses. The overlapping terms between Neurosynth (*Yarkoni et al., 2011*) and Cognitive Atlas (*Poldrack et al., 2011*) corpuses used in the reported analyses are listed below.

• Supplementary file 3. List of time-series features corresponding to PC1. The complete list of features (ranked by loading), their definitions, correlations and *p*-values for PC1 is presented in machine-readable format.

• Supplementary file 4. List of time-series features corresponding to PC2. The complete list of features (ranked by loading), their definitions, correlations and *p*-values for PC2 is presented in machine-readable format.

• Transparent reporting form

## Data availability

All data used in this study is publicly available. Detailed information about the datasets is available in the manuscript.

The following previously published datasets were used:

| Author(s) | Year | Dataset title | Dataset URL | Database and Identifier |
|---|---|---|---|---|
| Van Essen DC, Smith SM, Barch DM, Behrens TE, Yacoub E, Ugurbil K, WU-Minn HCP Consortium | 2013 | Human Connectome Project (HCP) | https://www.humanconnectome.org/study/hcp-young-adult | ConnectomeDB, 10.1016/j.neuroimage.2013.05.041 |
| Gordon EM, Laumann TO, Gilmore AW, Newbold DJ, | 2017 | Midnight Scan Club (MSC) | https://openneuro.org/datasets/ds000224/versions/1.0.0 | OpenfMRI database, ds000224 |

| | | | | | |
|---|---|---|---|---|---|
| Greene DJ, Berg JJ, Ortega M, Hoyt-Drazen C, Gratton C, Sun H, Hampton JM, Coalson RS, Nguyen AL, McDermott KB, Shimony JS, Snyder AZ, Schlaggar BL, Petersen SE, Nelson SM, Dosenbach NUF | | | | | |
| Hawrylycz MJ, Lein ES, Guillozet-Bongaarts AL, Shen EH, Ng L, Miller JA, van de Lagemaat LN, Smith KA, Ebbert A, Riley ZL, Abajian C, Beckmann CF, Bernard A, Bertagnolli D, Boe AF, Cartagena PM, Chakravarty MM, Chapin M, Chong J, Dalley RA, David DB, Dang C, Datta S, Dee N, Dolbeare TA, Faber V, Feng D, Fowler DR, Goldy J, Gregor BW, Haradon Z, Haynor DR, Hohmann JG, Horvath S, Howard RE, Jeromin A, Jochim JM, Kinnunen M, Lau C, Lazarz ET, Lee C, Lemon TA, Li L, Li Y, Morris JA, Overly CC, Parker PD, Parry SE, Reding M, Royall JJ, Schulkin J, Sequeira PA, Slaughterbeck CR, Smith SC, Sodt AJ, Sunkin SM, Swanson BE, Vawter MP, Williams D, Wohnoutka P, Zielke HR, Geschwind DH, Hof PR, Smith SM, Koch C, Grant SGN, Jones AR | 2012 | Allen Institute Human Brain Atlas (AHBA) | https://human.brain-map.org | Allen Institute Human Brain Atlas (AHBA), 10.1038/nn.4171 |

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
