## [Decision Letter]

**Acceptance summary:**

We believe that your work contributes novel insights into functional brain organization.

**Decision letter after peer review:**

Thank you for submitting your article "Topographic gradients of intrinsic dynamics across neocortex" for consideration by *eLife*. Your article has been reviewed by three peer reviewers, including Lucina Q Uddin as the Reviewing Editor and Reviewer #1, and the evaluation has been overseen by Chris Baker as the Senior Editor. The following individual involved in review of your submission has agreed to reveal their identity: Maxwell Bertolero (Reviewer #2).

The reviewers have discussed the reviews with one another and the Reviewing Editor has drafted this decision to help you prepare a revised submission.

Summary:

This manuscript describes a study that examines topographic gradients of intrinsic dynamics in the human brain. The authors identify two gradients with distinct temporal compositions and gene expression, myelin, cortical thickness, network embedding, and functional activation properties. The reviewers agreed that the reproducibility analyses (replication on independent datasets, use of two different parcellation schemes), and comparisons of data with and without (grey-matter signal regression) are particularly compelling. The similarities in temporal features between distinct brain regions revealed a ventromedial-dorsolateral (PC1) and a unimodal-transmodal (PC2) topographic gradients. Interestingly, analyses suggest a weak correlation between "temporal profile similarities" and functional connectivity. Broadly, these initial set of results suggest that the organization of spontaneous BOLD signal fluctuations link to both the geometrical and topological embedding of regions in macroscopic networks. The second set of analyses suggest that various measures of signal autocorrelations link to PC1 while measures of distribution shape (dynamic range) link to PC2. Moreover, the detected gradients are correlated with the spatial distribution of gene expression, cortical thickness, T1w/T2w, and functional hierarchy in an opposite fashion. The third set of analyses focus on linking topographic patterns in spontaneous fMRI signals with probability maps of task-based fMRI signals. The output of this analysis reveals that PC1 recapitulates a cognitive-affective axis, whereas the PC2 link to a sensory-cognitive axis. Overall, the reviewers were all in agreement regarding the execution, significance and approach of the study.

Essential revisions:

1) The overall narrative and rationale for the approach is missing. Some additional comments and suggestions regarding conceptual framing are below:

a) The Introduction as it is written includes a great deal of jargon that should be unpacked. In particular, the terms "contextual information", "network embedding", "spectral power", and "nonlinear dynamic models" should be defined clearly as they are introduced.

b) Cognitive ontologies come out of the blue in the Results, and are not mentioned at all in the Introduction. Similarly, the significance of gene expression and its relationship to network structure is not at all discussed in the Introduction. Again in the Discussion, the rationale, significance, and relevance of the gene expression findings are missing.

c) The paper provides numerous "confirmatory" findings, but after reading the paper a few times, I am left wondering what new fundamental knowledge the study provides. We know that fluctuations in spontaneous fMRI signal are patterned across the cortex, and that such patterns link to genetics, microstructural gradients and macroscopic functional and structural patterns. Perhaps the most novel/unexpected result is that biologically meaningful time-series features are not associated with functional connectivity. However, this result is only superficially discussed. The authors should make an extra effort to highlight better how the paper confirms (still a vital endeavour) and extends existing knowledge.

d) I am confused about the use of the term "dynamic". The adopted measures have little to do with dynamic measures of neural synchronization per se; they are more summary measure of these dynamics. The authors should be careful in their terminology and discuss the results for what they are (i.e., summary measures of spontaneous changes in fMRI signal).

2) Points of clarification and queries regarding analyses are below:

a) It seems that temporal features (autocorrelation, variance, spectral power, etc) are all considered together in the *hctsa* analysis. How do the authors account for the fact that these features are not necessarily independent? In other words, does the analysis consider the inter-relationships between these individual features?

b) Perhaps consider including a comparison evolutionary expansion or the participation coefficient (PC2 is likely anti-correlated with these).

c) The statistics reporting needs work. Please report all degrees of freedom. Also, please either report the results in the figure that shows the data, or in the text. In many places, the statistics are divided up, and that makes it hard to read. Also, there seem to be many points where a t-test was done, but only a p-value, not a t-value, is reported. Just saying p is close to zero is not informative if I don't know the DoF and the t value. For each finding, report the test type, the test value, the p value (with DoF), and then reference the panel. Or just reference the panel and put those stats in it. Or both! Also, don't put a * if you are not going to explain what it means.

d) Several temporal features do not seem orthogonal (e.g., power spectrum and autocorrelations). I may have missed this information but, were redundant features removed to obtain a balanced set of features describing the resting-state fMRI signal? If similar features were not removed, results might have been biased towards aspects of rest fMRI signals that are over-represented. The Results section suggests that this may be the case.

e) Patterns of structural connectivity are known to be linked to spontaneous fluctuations in functional connectivity. I suggest extending these results by providing a quantification of the shared/distinct contributions of functional network topology and anatomical factors to temporal profile similarity.

f) Could the authors consider using an estimate of functional connectivity that accounts for shared signal (e.g., partial correlation, multiple regression)? I think this would shed light on the relationships identified in the paper and how much can be related to likely false positives found in correlation FC.

g) There is little discussion regarding why the variables (e.g., Figure 4) should (or should not) be related to each other (see also reviewer comment a). Given that many of these measures are inter-related, it is not entirely clear to me if they are meaningful. An example of this is the positive/negative relationships observed in Figure 4: Is this bound to happen because of the relationship between PC1 and PC2? Is this an interesting relationship, or just a statistical consequence of PCA?

---

## [Author Response]

Essential revisions:1) The overall narrative and rationale for the approach is missing. Some additional comments and suggestions regarding conceptual framing are below:a) The Introduction as it is written includes a great deal of jargon that should be unpacked. In particular, the terms "contextual information", "network embedding", "spectral power", and "nonlinear dynamic models" should be defined clearly as they are introduced.

We have now clarified these definitions and have modified the manuscript accordingly. Specific changes are explained below (Introduction):

Re: “contextual information”:

“The primary functional consequence of this hierarchy of timescales is thought to be a hierarchy of temporal receptive windows: time windows in which a newly arriving stimulus will modify processing of previously presented (i.e. contextual) information (Hasson et al., 2008; Honey et al., 2012; Baldassano et al., 2017; Huntenburg et al., 2018; Chaudhuri et al., 2015; Chien and Honey, 2020). […] Altogether, previous work highlights a hierarchy of a small number of manually selected specific time-series features, but it is possible that different types of local computations manifest as different organizational gradients.”

Re: “network embedding”:

“The signal variability of brain areas, measured in terms of standard deviations or temporal entropy, is closely related to their structural and functional connectivity profiles (i.e. network embedding) (Misic et al., 2011; Burzynska et al., 2013; Garrett et al., 2017; Shafiei et al., 2019).”

Re: “Spectral power”:

“A significant limitation is that conventional computational analysis is based on specific, manually selected time-series features, such as the decay of the autocorrelation function, bands of the Fourier power spectrum, or signal variance.”

Re: “nonlinear dynamic models”:

“Finally, in computational models of structurally coupled neuronal populations (neural mass and neural field models; Breakspear, 2017), highly interconnected hubs exhibit slower dynamic fluctuations, while sensory areas exhibit fast fluctuating neural activity (Gollo et al., 2015). Indeed, these models offer better fits to empirical functional connectivity if they assume heterogeneous local dynamics that follow a unimodal—transmodal gradient (Cocchi et al., 2016; Demirtas et al., 2019; Wang et al., 2019; Deco et al., 2020).”

b) Cognitive ontologies come out of the blue in the Results, and are not mentioned at all in the Introduction. Similarly, the significance of gene expression and its relationship to network structure is not at all discussed in the Introduction. Again in the Discussion, the rationale, significance, and relevance of the gene expression findings are missing.

We agree, and have modified the manuscript accordingly. Specific changes are explained below:

Re: cognitive ontologies:

We first introduce the rationale for the analysis; namely, that spatial variation in circuit structure may shape circuit dynamics and computations, manifesting as patterns of regional functional specialization (Introduction):

“Altogether, multiple lines of evidence suggest that local computations may reflect systematic variation in microscale properties and macroscale network embedding, manifesting as diverse time-series features of regional neural activity. How molecular, cellular and connectomic architecture precisely shapes temporal dynamics, and ultimately, cortical patterns of functional specialization, is poorly understood.”

We then explicitly lay out how we will address this question: by mapping time-series features to reverse-inference meta-analytic maps (Introduction):

“Finally, we map time-series features to a meta-analytic atlas of cognitive ontologies to investigate how temporal dynamics shape regional functional specialization. […] These spatial variations in intrinsic dynamics ultimately manifest as patterns of distinct psychological functions.”

Re: gene expression

We have modified the Introduction to first intuitively set the idea that gene expression may convey information about cellular structure and function, and may therefore be related to intrinsic dynamics (Introduction):

“For a neuronal population, the confluence of local micro-architectural properties and global connectivity shapes both the generation of local rhythms, as well as its propensity to communicate with other populations. […] These micro-architectural properties – increasingly measured directly from histology or inferred from other measurements, such as microarray gene expression – provide a unique opportunity to relate circuit architecture to temporal dynamics and computation.”

We have also substantially revised the Discussion to explain the significance of these findings. We discuss this change in more detail in our response to comment #2, point g, but have included the new text here as well (Discussion section):

“Applying a data-driven feature extraction method to high-resolution BOLD fMRI, we decompose regional signals into two intrinsic modes, with distinct topographic organization and time-series features. […] Taken together, we find evidence that molecular and cellular properties (gene expression PC1) relate to regional autocorrelation, while micro-circuit properties (T1w/T2w, cortical thickness) and macroscale network embedding (principal functional gradient) relate to regional dynamic range.”

c) The paper provides numerous "confirmatory" findings, but after reading the paper a few times, I am left wondering what new fundamental knowledge the study provides. We know that fluctuations in spontaneous fMRI signal are patterned across the cortex, and that such patterns link to genetics, microstructural gradients and macroscopic functional and structural patterns. Perhaps the most novel/unexpected result is that biologically meaningful time-series features are not associated with functional connectivity. However, this result is only superficially discussed. The authors should make an extra effort to highlight better how the paper confirms (still a vital endeavour) and extends existing knowledge.

We concur that the original manuscript did not fully highlight the novelty of the findings and conceptual advances made by the study. In the revised manuscript, we emphasize 5 key contributions:

1) Comprehensive time-series phenotyping. The majority of the field focuses on specific univariate time-series features-of-interest, such as measures of intrinsic time scale, variance, or spectral power. Our approach is the first to comprehensively benchmark the entire dynamic profile of the brain, by near-exhaustively estimating 6000+ features from the wider time-series literature. This represents a significant conceptual advance, as it offers a fundamentally new and entirely data-driven method to quantify fMRI dynamics. Focusing on univariate time-series features is a powerful but under-utilized direction to understand neural dynamics.

A salient example is that when multiple time-series features are taken into account, we observe two prominent yet distinct modes of intrinsic dynamics, with different topographies and dynamical signatures. Consider the spatial variation of BOLD autocorrelation; multiple accounts posit that the unimodal-transmodal hierarchy is expressed as regional differences in autocorrelation or intrinsic timescale (Hasson et al., 2008; Murray et al., 2014; Honey et al., 2012; Ito et al., 2020). Yet the present results show that this unimodal-transmodal hierarchy is more closely related to regional differences in dynamic range, whereas regional differences in autocorrelation display a different spatial pattern. Indeed, the dynamic range pattern (PC2) more closely corresponds to multiple measures of microarchitecture than the autocorrelation pattern (PC1). By comprehensively studying multiple time-series features and multiple anatomical features, the present report shows that — what is typically assumed to be a single hierarchy — is actually two hierarchies with distinct structural, dynamic and psychological attributes. To clarify this contribution, we have added the following text to the Discussion (Discussion section):

“Our results demonstrate that regional haemodynamic activity, often overlooked in favour of electrophysiological measurements with greater temporal resolution, possesses a rich dynamic signature (Garrett et al., 2013; Uddin, 2020; Preti et al., 2017; Lurie et al., 2020; Li et al., 2019; Bolt et al., 2018). […] As we discuss below, feature-based time-series phenotyping offers a powerful, fundamentally new and entirely data-driven method to quantify and articulate neural dynamics.”

2) Systematic mapping of time-series features to architectural features. A related point is that most previous literature has sought to relate small manually selected sets of time-series features-of-interest to small manually selected sets of structural features-of-interest. The present work integrates comprehensive dynamic profiles to architectural features across multiple scales, including molecular, cellular, laminar and connectomic features. To our knowledge, there exist no reports or records of these relationships, so the present findings will be a touchstone for future research in this domain. To clarify this contribution, we have added the following paragraph to the Discussion (Discussion section): (see also our response to comment #2, point g)

“Applying a data-driven feature extraction method to high-resolution BOLD fMRI, we decompose regional signals into two intrinsic modes, with distinct topographic organization and time-series features[…] How these patterns are related to underlying cell types and subcortical afferent input – in particular, thalamocortical feedback – is an important ongoing question (Abeysuriya et al., 2015; Garrett et al., 2018; Shine et al., 2019; Wang et al., 2019; Muller et al., 2020; Paquola et al., 2020).”

3) Linking structure, dynamics, and cognition. We show that topographic variations in microcircuitry and connectomic embedding yield variations in intrinsic dynamics and, importantly, may explain regional differences in functional specialization. Specifically, we show that patterns of intrinsic dynamics are concomitant with two dominant axes of functional activation, derived from Neurosynth. Thus, the present findings conceptually link structural, temporal and neurocognitive architectures. To clarify this contribution, we have added the following paragraph to the Discussion (Discussion section):

“Importantly, the two patterns are related to two dominant axes of meta-analytic functional activation. […] Collectively, these results provide evidence that local computations reflect systematic variation in multiple anatomical circuit properties, and can be measured as unique temporal signatures in regional activity and patterns of functional specialization.”

4) Re-conceptualizing functional connectivity. At present, functional relationships are typically conceptualized as bivariate relationships among regional time-series, with the implicit assumption that if two regions display coherent dynamics, they may be engaged in a common function. Here we show that, by deconstructing time-series into 6000+ features, it is possible to operationalize inter-regional relationships as similarity of their dynamic properties. Importantly, we show that traditional, correlation-based FC is only weakly associated with temporal profile similarity. In other words, the traditional conceptualization of functional connectivity misses out on an important set of inter-regional relationships. Namely, two regions may display identical time-series profiles, suggesting common circuit dynamics and function, but unless they also display time-locked activity, current methods would miss out on this potentially biologically meaningful inter-regional relationship. We have clarified this contribution in the revised manuscript (Discussion section):

“Indeed, we find that two regions are more likely to display similar intrinsic dynamics if they are anatomically connected and if they are part of the same functional community, suggesting that network organization and local intrinsic dynamics are intertwined (Gollo et al., 2015; Cocchi et al., 2016). […] Namely, two regions may display identical time-series profiles, suggesting common circuit dynamics and function, but unless they also display time-locked activity, current methods would miss out on this potentially biologically meaningful inter-regional relationship.”

5) Extensive robustness testing. The results are subjected to numerous sensitivity and replication analyses, which demonstrate that the findings are consistent across:

a) Group- and individual-level

b) With and without grey-matter signal regression

c) Parcellation (anatomical and functional) and parcellation resolution

d) Out-of-sample validation in held-out HCP data and MSC data

We have added the following sentence to the Discussion (Discussion section):

“These findings are robust against a wide range of methodological choices and were validated in two held-out samples.”

d) I am confused about the use of the term "dynamic". The adopted measures have little to do with dynamic measures of neural synchronization per se; they are more summary measure of these dynamics. The authors should be careful in their terminology and discuss the results for what they are (i.e., summary measures of spontaneous changes in fMRI signal).

We have added the following sentence in the Introduction (at the first mention of the term “dynamic”) to clarify that the time-series features are summary measures of local neural activity (Introduction):

“Here we comprehensively chart summary features of spontaneous BOLD signals across the cerebral cortex (hereafter referred to as “intrinsic dynamics”), mapping temporal organization to structural organization.”

We have also changed the term “dynamic profile” to “time-series profile” throughout the manuscript.

2) Points of clarification and queries regarding analyses are below:a) It seems that temporal features (autocorrelation, variance, spectral power, etc) are all considered together in the hctsa analysis. How do the authors account for the fact that these features are not necessarily independent? In other words, does the analysis consider the inter-relationships between these individual features?

This is precisely why we applied PCA to the *hctsa* output – to identify groups of correlated features. The analysis produces 6,441 time-series features that are nearly-exhaustive but some may potentially be correlated. As a result, we sought to find linear combinations of features that explain maximal variance (principal components) and that span different conceptual types of time-series properties.

To clarify this rationale, we have added the following passage to the revised manuscript (Results section, “Two distinct spatial gradients of intrinsic dynamics” subsection):

“The *hctsa* library generates 6,441 time-series features, with the aim of being comprehensive in coverage across scientific time-series analysis algorithms and, as a result, contains groups of features with correlated outputs (Fulcher et al., 2013). We therefore sought to identify groups of correlated features that explain maximal variance and that span different conceptual types of time-series properties.”

To provide an intuitive sense of whether top-loading features in each component (full list in Supplementary files 3, 4) display fMRI-specific patterning, we additionally plotted two high-loading, representative features for each component in Figure 3C (top). To build intuition about what each component reflects about regional signals, we selected three regional time-series from one participant based on their lag-1 autocorrelation and kurtosis (Figure 3C; circles on the brain surface: pink = 5^th^ percentile, green = 50^th^ percentile, purple = 95^th^ percentile). For the former, going from low-ranked to high-ranked regions results in a slowing down of BOLD fluctuations. For the latter, going from low-ranked to high-ranked regions results in increasingly heavier symmetric tails of the signal amplitude distributions.

b) Perhaps consider including a comparison evolutionary expansion or the participation coefficient (PC2 is likely anti-correlated with these).

We thank the reviewers for their suggestion. We have now added an analysis (Figure 4—figure supplement 2) comparing the two maps with the PCA results. The evolutionary expansion map (Hill et al., 2010; Baum et al., 2020) was obtained through https://github.com/PennLINC/Brain_Organization and parcellated into 400 cortical areas using the Schaefer parcellation (Schaefer et al., 2018). Regional weighted participation coefficients (Brain Connectivity Toolbox; Rubinov and Sporns, 2010) were estimated from functional connectivity graphs with respect to the 7-network partition of intrinsic networks (Yeo et al., 2011; Schaefer et al., 2018). We then compared the maps to the PC1 and PC2 brain score patterns. The evolutionary expansion map was significantly correlated with PC2 topography (*r*_s_ = 0.52, *p*_spin_ = 0.0002), but not PC1 topography (*r*_s_ = 0.01, *p*_spin_ = 0.95). The regional participation coefficient map was not significantly correlated with either PC1 or PC2 topography (*r*_s_ = -0.14, *p*_spin_ = 0.16; and *r*_s_ = -0.10, *p*_spin_ = 0.32; respectively).

We have also added the following passage to the revised manuscript (“Results section, “Intrinsic dynamics reflect microscale and macroscale hierarchies” subsection):

“For completeness, we also tested associations with two maps that were previously related to cortical hierarchies: evolutionary expansion (indexing enlargement of cortical areas in the human relative to the macaque) (Hill et al., 2010; Baum et al., 2020) and node-wise functional participation coefficient (indexing the diversity of a node's links) (Bertolero et al., 2017; Baum et al., 2020). PC2 is significantly correlated with evolutionary expansion (*r*_s_ = 0.52, *p*_spin_ = 0.0002), but neither component is correlated with participation coefficient (Figure 4—figure supplement 2).”

c) The statistics reporting needs work. Please report all degrees of freedom. Also, please either report the results in the figure that shows the data, or in the text. In many places, the statistics are divided up, and that makes it hard to read. Also, there seem to be many points where a t-test was done, but only a p-value, not a t-value, is reported. Just saying p is close to zero is not informative if I don't know the DoF and the t value. For each finding, report the test type, the test value, the p value (with DoF), and then reference the panel. Or just reference the panel and put those stats in it. Or both! Also, don't put a * if you are not going to explain what it means.

Thank you for pointing this out. We have added the following explanation to the Figure 2 legend to denote what asterisks represent:

“(c, d) Regional time-series features are compared between pairs of cortical areas using their structural and functional connectivity profiles. […] For functional networks, statistical significance of the difference of the mean temporal profile similarity of within and between intrinsic networks is also assessed against a null distribution of differences generated by spatial autocorrelation-preserving label permutation (“spin tests”; Alexander-Bloch et al., 2018) (d, right-most panel).”

We have also included complete information for all *t*-tests, including degrees of freedom and exact *t*-values (Results section, “Inter-regional temporal profile similarity reflects network geometry and topology” subsection):

Structurally connected vs. not connected temporal profile similarity (Figure 2C):

*t*(79,798) = 40.234; *p* ≅ 0

for distance regressed temporal profile similarity: *t*(79,798) = 9.916; *p* ≅ 0

Within vs. between intrinsic network temporal profile similarity (Figure 2D):

*t*(79,798) = 61.093; *p* ≅ 0

for distance regressed temporal profile similarity: *t*(79,798) = 47.112; *p* ≅ 0

We note that, since inference is performed across a large number of edges, there are correspondingly large degrees of freedom. To ensure that the results are not simply an artefact of sample size, we additionally used nonparametric tests, for which we show null distributions and empirical values in the right-most panels of the figure. Specifically, we used rewired networks for estimating similarity of structurally-connected and -unconnected node pairs, and we used spatial autocorrelation-preserving spin tests to assess the similarity of within- and between-functional network node pairs (both panels shown in Figure 2C, D).

d) Several temporal features do not seem orthogonal (e.g., power spectrum and autocorrelations). I may have missed this information but, were redundant features removed to obtain a balanced set of features describing the resting-state fMRI signal? If similar features were not removed, results might have been biased towards aspects of rest fMRI signals that are over-represented. The Results section suggests that this may be the case.

No features were removed, as the goal of the analysis was to comprehensively characterize the temporal organization of neural activity, without any pre-selection of features. As we outline below, this approach opens fundamentally new ways to analyze neural activity, but potential correlations among features must be explicitly considered and analyzed.

Although some specific *hctsa* measures attempt to capture similar properties, each one uses a different method and/or parameter settings, yielding 6,441 unique features. At the same time, features could be correlated with each other (e.g. outputs of autocorrelation measures) (Fulcher et al., 2013). This is precisely why we applied PCA to the *hctsa* output – to identify groups of correlated features. Specifically, we sought to find linear combinations of features that explain maximal variance (principal components) and that span different conceptual types of time-series properties. This approach allows us to discover connections between time-series properties that have different names and come from different literatures, but behave similarly. To clarify this rationale, we have added the following passage to the revised manuscript (Results section, “Two distinct spatial gradients of intrinsic dynamics” subsection):

“The *hctsa* library generates 6,441 time-series features, with the aim of being comprehensive in coverage across scientific time-series analysis algorithms and, as a result, contains groups of features with correlated outputs (Fulcher et al., 2013). We therefore sought to identify groups of correlated features that explain maximal variance and that span different conceptual types of time-series properties.”

Analyzing all features allowed us to map the temporal organization of neural activity across the brain. However, the downside of using all possible features is that some will be correlated with each other. Although we applied PCA to guard against this, the solutions may be biased towards the more numerous features. To acknowledge this limitation, we have added the following text to the Discussion (Discussion section):

“Fourth, the analysis included all features from *hctsa*, potentially biasing results towards specific properties of BOLD signals. […] This may obscure the contribution of under-represented feature classes, and should be investigated further in future work.”

e) Patterns of structural connectivity are known to be linked to spontaneous fluctuations in functional connectivity. I suggest extending these results by providing a quantification of the shared/distinct contributions of functional network topology and anatomical factors to temporal profile similarity.

Inter-regional temporal profile similarity could potentially be driven by inter-regional (a) distance, (b) structural connectivity, (c) functional connectivity. To quantify the distinct contributions of each factor, we first constructed a multiple linear regression with distance, structural connectivity and functional connectivity as independent variables and temporal profile similarity as the dependent variable. Observations were edges or node pairs. The analysis was restricted to structurally connected node pairs (k = 4,950 unique edges). We then applied Dominance Analysis, a method for assessing the relative importance of predictors (Budescu, 1993; Azen and Budescu, 2003; https://github.com/dominance-analysis/dominance-analysis).

Briefly, the analysis measures relative importance in a series of pairwise comparisons, where two predictors are compared in the context of all possible models that contain some subset of the other predictors (i.e. for *p* predictors we have 2*^p^*-1 models). The incremental *R*^2^ contribution of each predictor to the subset model of all other predictors is then computed. The Total Dominance of each predictor is summarized as the additional contribution of a predictor to all subset models.

The results of the analysis are shown in Supplementary file 1. The total *R*^2^ is 0.28 for the complete model, with the following relative importance (last column): distance – 56%, FC – 23.6%, SC – 20.4%. In other words, distance contributes the most to temporal profile similarity, while structural and functional connectivity make distinct but approximately even contributions.

We have also added the following passage (Results section, “Inter-regional temporal profile similarity reflects network geometry and topology” subsection):

“As a final step, we sought to assess the distinct contributions of distance, structural connectivity and functional connectivity to temporal profile similarity. Dominance analysis revealed the relative importance of each predictor (collective R^2^ = 0.28; distance = 56%, structural connectivity = 20.4%, functional connectivity = 23.6%; Supplementary file 1), suggesting that distance contributes the most to temporal profile similarity, while structural and functional connectivity make distinct but approximately even contributions (Budescu, 1993; Azen and Budescu, 2003; https://github.com/dominance-analysis/dominance-analysis).”

f) Could the authors consider using an estimate of functional connectivity that accounts for shared signal (e.g., partial correlation, multiple regression)? I think this would shed light on the relationships identified in the paper and how much can be related to likely false positives found in correlation FC.

We have repeated all FC-based analyses using partial correlations, as implemented in *Nilearn* (Abraham et al., 2014). The results are virtually identical to before. Namely, there is a weak relationship between functional connectivity and temporal profile similarity.

We have included the new results as a Figure 2—figure supplement 2 and have added the following sentences to the revised manuscript to refer to this figure (Results section, “Inter-regional temporal profile similarity reflects network geometry and topology” subsection):

“The results are consistent when functional connectivity is estimated using partial correlations (Figure 2—figure supplement 2).”

g) There is little discussion regarding why the variables (e.g., Figure 4) should (or should not) be related to each other (see also reviewer comment a). Given that many of these measures are inter-related, it is not entirely clear to me if they are meaningful. An example of this is the positive/negative relationships observed in Figure 4: Is this bound to happen because of the relationship between PC1 and PC2? Is this an interesting relationship, or just a statistical consequence of PCA?

Why are these relationships significant?

Existing work has highlighted dominant cortical hierarchies but perhaps different types of local computations manifest in different organizational gradients. Our goal was to systematically compare temporal and micro-architectural features. By comprehensively mapping time-series features to other well-studied maps (gene expression, functional gradient, intracortical myelin, cortical thickness, evolutionary expansion), our results can be readily contextualized by other researchers studying brain organization from molecular, cellular or macroscale perspectives. Throughout the analysis, we apply conservative autocorrelation-preserving spin tests to assess associations.

Specifically, we find two dominant but distinct temporal modes that are expressed over the cortex, one indexing signal autocorrelation and the other indexing dynamic range. The first is closely associated with gene expression PC1 (itself closely related to cell type composition, synaptic physiology and cortical cytoarchitecture; Burt et al., 2018), suggesting a molecular and cellular basis for regional differences in temporal autocorrelation. The second is closely associated with the principal functional gradient, as well as with intracortical myelin and cortical thickness, suggesting that the dynamic range of BOLD signals is related to regional variation in macroscale circuit organization. Taken together, we find evidence that molecular and cellular properties (gene expression PC1) relate to regional autocorrelation, while micro-circuit properties (T1w/T2w, cortical thickness) and macroscale network embedding (principal functional gradient) relate to regional dynamic range.

To clarify these relationships, we have included the following passage to the Discussion (Discussion section):

“Applying a data-driven feature extraction method to high-resolution BOLD fMRI, we decompose regional signals into two intrinsic modes, with distinct topographic organization and time-series features. […] How these patterns are related to underlying cell types and subcortical afferent input — in particular, thalamocortical feedback — is an important ongoing question (Abeysuriya et al., 2015; Garrett et al., 2018; Shine et al., 2019; Wang et al., 2019; Muller et al., 2020; Paquola et al., 2020).”

Interestingly, the present results add subtlety to the emerging literature on multimodal cortical gradients. As the reviewers point out, many reports suggest that these gradients are inter-related, with most assuming a unimodal-transmodal hierarchy. In the time-series dynamics literature, this is partly due to the fact that researchers often study specific time-series features of interest. However, the present results show that when multiple time-series features are taken into account, we observe two prominent yet distinct modes of intrinsic dynamics, with different topographies and dynamical signatures. A salient example is the distribution of regional autocorrelation; multiple accounts posit that the unimodal-transmodal hierarchy is expressed as regional differences in autocorrelation or intrinsic timescale. Yet the present results show that the unimodal-transmodal hierarchy is more closely related to regional differences in dynamic range, whereas regional differences in autocorrelation follow a different topographic distribution. By comprehensively studying multiple time-series features and multiple anatomical features, the present report shows that, what is typically assumed to be a single hierarchy, is actually two hierarchies with distinct structural, dynamic and psychological attributes.

Is this a consequence of PCA?

In some instances, PC1 and PC2 scores have correlations with different signs. This is not an artifact of the principal component analysis. Namely, the signs of PC weights are arbitrary; multiplication by -1 (corresponding to axis reflection), yields components with equal effect size (variance accounted for) (McIntosh and Misic, 2013). The only constraint in the analysis is that the eigenvectors (PC weights) are mutually orthogonal. Thus, there is no a priori reason for PC scores to be positively correlated, negatively correlated, or anticorrelated with a third measure of interest, such as gene expression, the principal gradient, etc.

Reference:

McIntosh, A. R., and Mišić, B. (2013). Multivariate statistical analyses for neuroimaging data. Annual review of psychology, 64, 499-525.